# Resolving the mesospheric nighttime 4.3 $\mu$m emission puzzle: Comparison of the $CO_2(\nu_3)$ and OH($v$) emission models with space and ground based observations

Peter A. Panka[1,2], Alexander A. Kutepov[2,3], Konstantinos S. Kalogerakis[4], Diego Janches[2], James M. Russell[5], Ladislav Rezac[6], Artem G. Feofilov[7], Martin G. Mlynczak[8], and Erdal Yiğit[1]

[1]Department of Physics and Astronomy, George Mason University, Fairfax, Virginia, USA.
[2]NASA Goddard Space Flight Center, Greenbelt, MD, USA.
[3]The Catholic University of America, Washington, DC, USA.
[4]Center for Geospace Studies, SRI International, Menlo Park, California, USA.
[5]Center for Atmospheric Sciences, Hampton University, Hampton, VA, USA.
[6]Max Planck Institute for Solar System Research, Göttingen, Germany.
[7]Laboratoire de Météorologie Dynamique/IPSL/FX-Conseil, CNRS, Ecole Polytechnique, Université Paris-Saclay, 91128 Palaiseau, France.
[8]NASA Langley Research Center, Hampton, Virginia, USA.

*Correspondence to:* P. A. Panka (ppanka@masonlive.gmu.edu)

**Abstract. In the 1970s, the mechanism of vibrational energy transfer from chemically produced OH($\nu$) in the nighttime mesosphere to the $CO_2(\nu_3)$ vibration, $OH(\nu) \Rightarrow N_2(\nu) \Rightarrow CO_2(\nu_3)$ was proposed. In later studies it was shown that this "direct" mechanism for simulated nighttime 4.3 $\mu$m emissions of the mesosphere is not sufficient to explain space observations. In order to better match these observations, an additional enhancement is needed that would be equivalent**

**to the production of 2.8-3 $N_2(1)$ molecules instead of one $N_2(1)$ molecule in each quenching reaction of $OH(\nu)+N_2(0)$. Recently a new "indirect" channel of the OH($\nu$) energy transfer to $N_2(\nu)$ vibrations, $OH(\nu) \Rightarrow O(^1D) \Rightarrow N_2(\nu)$, was suggested and then confirmed in a laboratory experiment, where its rate for $OH(\nu=9)+O(^3P)$ was measured. We studied in detail the impact of the "direct" and "indirect" mechanisms on $CO_2(\nu_3)$ and OH(v) vibrational level populations and emissions. We also compared our calculations with (a) the SABER/TIMED nighttime 4.3 $\mu$m $CO_2$ and OH 1.6 and**

**2.0 $\mu$m limb radiances of the MLT and (b) with ground and space observations of OH($\nu$) densities in the nighttime mesosphere. We found that the new "indirect" channel provides a strong enhancement of the 4.3 $\mu$m $CO_2$ emission, which is comparable to that obtained with the "direct" mechanism alone but assuming an efficiency that is 3 times higher. Both models match well with SABER nighttime 4.3 $\mu$m radiances. The model based on the "indirect" channel is also in good agreement with both SABER limb OH emission observations and ground and space measurements of**

**OH(v) densities. This is, however, not true for the model which relies on the "direct" mechanism alone. This mismatch is caused by the lack of an efficient redistribution of the OH($\nu$) energy from higher vibrational levels emitting at 2.0 $\mu$m to lower levels emitting at 1.6 $\mu$m. In contrast, this new "indirect" mechanism efficiently removes at least 5 quanta in each $OH(\nu \geq 5)+O(^3P)$ collision and provides the OH($\nu$) distributions, which agree with both SABER limb OH emission observations and ground and space based OH(v) density measurements. This analysis suggests that the important**

mechanism of the OH($v$) vibrational energy relaxation in the nighttime MLT, which was missing in the emission models of this atmospheric layer, has been finally identified.

# 1  Introduction

A detailed study of nighttime 4.3 $\mu$m emissions was conducted in by López-Puertas et al. (2004) aimed at determining the dominant mechanisms of exciting $CO_2(\nu_3)$, where $\nu_3$ is the asymmetric stretch mode that emits 4.3 $\mu$m radiation. The nighttime measurements of SABER channels 7 (4.3 $\mu$m), 8 (2.0 $\mu$m), and 9 (1.6 $\mu$m) for geomagnetically quiet conditions were analyzed, where channels 8 and 9 are sensitive to the OH ($\nu \leq 9$) overtone radiation from levels $\nu = 8$–9 and $\nu = 3$–5, respectively. López-Puertas et al. (2004) showed a positive correlation between 4.3 $\mu$m and both OH channel radiances at a tangent height of 85 km. This correlation was associated with the transfer (Kumer et al., 1978) of energy of the vibrationally excited OH($\nu$) produced in the chemical reaction

$$H + O_3 \rightarrow O_2 + OH(\nu \leq 9) \tag{R1}$$

first to $N_2(1)$

$$OH(\nu \leq 10) + N_2(0) \leftrightarrow OH(\nu - 1) + N_2(1), \tag{R2}$$

and then further to $CO_2(\nu_3)$ vibrations

$$N_2(1) + CO_2(0) \leftrightarrow N_2(0) + CO_2(\nu_3) \tag{R3}$$

(hereafter "direct" mechanism). However, López-Puertas et al. (2004) showed that calculations based on the (Kumer et al., 1978) model do not allow reproducing the 4.3 $\mu$m radiances observed by SABER. Although accounting for energy transfer from OH($\nu$) did provide a substantial enhancement to 4.3 $\mu$m emission, a 40% difference between simulated and observed radiance remained (for the SABER scan 22, orbit 01264, 77°N, 03 Mar 2002, which was studied in detail) for altitudes above 70 km. In order to fit measurements these authors found that, on average, 2.8-3 $N_2(1)$ molecules (instead of Kumer's suggested value of 1) are needed to be produced after each quenching of OH($\nu$) molecule in reaction (R2). Alternative excitation mechanisms that were theorized to enhance the 4.3 $\mu$m radiance (i.e. via $O_2$ and direct energy transfer from OH to $CO_2$) were tested but found to be insignificant.

Recently, Sharma et al. (2015) suggested a new "indirect" mechanism of the OH vibrational energy transfer to $N_2$, i.e. $OH(\nu) \Rightarrow O(^1D) \Rightarrow N_2(\nu)$. Accounting for this mechanism, but only considering OH($\nu$=9), these authors performed simple model calculations to validate its potential for enhancing mesospheric nighttime 4.3 $\mu$m emission from $CO_2$. They reported a simulated radiance enhancement between 18-55% throughout the MLT. In a latest study, Kalogerakis et al. (2016) provided a definitive laboratory confirmation for the validity of this new mechanism and measured its rates for OH($\nu$=9)+O.

**We studied in detail the impact of "direct" and "indirect" mechanisms on the $CO_2(\nu_3)$ and OH(v) vibrational level populations and emissions and compared our calculations with (a) the SABER/TIMED nighttime 4.3 $\mu$m $CO_2$ and OH 1.6 and 2.0 $\mu$m limb radiances of MLT and (b) with the ground and space observations of the OH($\nu$) densities in night time mesosphere.** The study was performed for quiet (non-auroral) nighttime conditions to avoid accounting for interactions between charged particles and molecules, whose mechanisms still remain poorly understood.

## 2 Non-LTE Model

A non-LTE analysis was applied to $CO_2$ and OH using the non-LTE ALI-ARMS (Accelerated Lambda Iterations for Atmospheric Radiation and Molecular Spectra) code package (Kutepov et al. (1998), Gusev and Kutepov (2003), Feofilov and Kutepov (2012)), which is based on the Accelerated Lambda Iteration approach (Rybicki and Hummer, 1991).

Our $CO_2$ non-LTE model is described in detail by Feofilov and Kutepov (2012). We modified its nighttime version to account for the "direct" mechanism, reactions (R1-R3), in a way consistent with that of López-Puertas et al. (2004) and added the "indirect" mechanism of Sharma et al. (2015) and Kalogerakis et al. (2016) as described in detail below. Our OH non-LTE model resembles that of Xu et al. (2012).

### 2.1 New Mechanism of OH($v$) relaxation

Sharma et al. (2015) suggested an additional mechanism that contributes to the $CO_2(\nu_3)$ excitation at nighttime, and discussed in detail its available experimental and theoretical evidence. According to this mechanism, highly vibrationally excited OH($\nu$), produced by reaction (R1), rapidly loses several vibrational quanta in collisions with O($^3$P) through a fast, spin-allowed, vibration-to-electronic energy transfer process that produces O($^1$D)

$$OH(\nu \geq 5) + O(^3P) \leftrightarrow OH(0 \leq \nu' \leq \nu - 5) + O(^1D). \tag{R4}$$

Recently, Kalogerakis et al. (2016) have presented the first laboratory demonstration of this new OH($\nu$) + O($^3$P) relaxation pathway and measured its rate coefficient for $v = 9$.

    The production at nighttime of electronically excited O($^1$D) atoms in reaction (R4) triggers well known pumping mechanism of the 4.3 $\mu$m emission, which was studied in detail for daytime (Nebel et al. (1994), Edwards et al. (1996)). Here O($^1$D) atoms are first quenched by collisions with $N_2$ in a fast spin-forbidden energy transfer process

$$O(^1D) + N_2(0) \leftrightarrow O(^3P) + N_2(\nu), \tag{R5}$$

then $N_2(\nu)$ transfers its energy to ground state $N_2$ via a very fast single quantum VV process

$$N_2(\nu) + N_2(0) \leftrightarrow N_2(\nu - 1) + N_2(1),$$

leaving $N_2$ molecules with an average of 2.2 vibrational quanta, which is then followed by reaction (R3).

### 2.2 Collisional Rate Coefficients

We use, in our $CO_2$ non-LTE model, the same VT and VV collisional rate coefficients for the $CO_2$ lower vibrational levels as those of López-Puertas et al. (2004). However, a different scaling of these basic rates is applied for higher vibrational levels using the first-order perturbation theory as suggested by Shved et al. (1998).

    The reaction rate coefficients applied in this study for modeling OH($v$) relaxation transfer of OH($\nu$) vibrational energy to the $CO_2(\nu_3)$ mode are displayed in Table 1. The total chemical production rate of OH($\nu$) in reaction (R1) was taken from Sander

et al. (2011) and the associated branching ratios for $\nu$ were taken from Adler-Golden (1997). **We treat reaction (R2) both as a single (1Q, $\nu'$=1) and multi-quantum (MQ, $\nu'$=2 or 3) quenching process. We use the rate coefficient of this reaction (with associated branching ratios) taken from Table 1 of (Adler-Golden, 1997) and multiplied it by a low temperature factor of 1.4 (Lacoursière et al., 2003) for MLT regions.** The rate coefficient for reaction R3 was taken from (Shved et al., 1998).

Following Sharma et al. (2015) and Kalogerakis et al. (2016), the rate coefficient of reaction (R4) was taken $(2.3\pm1)\times10^{-10}$ $cm^3s^{-1}$ for temperatures near 200 K (with corresponding branching ratios for $v \geq 5$. Additionally, for OH($\nu <5$) collisions with O($^3$P), which are considered completely inelastic, we used the rate coefficient $5\times10^{-11}$ $cm^3s^{-1}$ from (Caridade et al., 2013). The rate coefficient for the reaction O($^1$D) + N$_2$(0) (R5 in Table 1) was taken from Sander et al. (2011) with accounting for the fact that 33% of the electronic energy is transferred to N$_2$ (Slanger and Black, 1974) producing, on average, 2.2 N$_2$ vibrational quanta. **For the reaction OH($\nu \leq$9) + O$_2$(0) (reaction (R6) in Table 1), we consider single and multi-quantum quenching, using rate coefficients with associated branching ratios taken from Table 1 and Table 3 of (Adler-Golden, 1997), respectively. Rate coefficients are scaled by a factor of 1.18 to account for MLT temperatures (Lacoursière et al. (2003), Thiebaud et al. (2010)). Lastly, reaction (R7) describes an alternative OH-O quenching mechanism which previous studies (López-Puertas et al. (2004), Adler-Golden (1997)) applied in their OH models, where atomic oxygen completely quenches ($\nu \to \nu$=0) OH($\nu$) upon collision. For this reaction, we took the vibrationally independent rate coefficient of $2.0\times10^{-10}$ $cm^3s^{-1}$ from Adler-Golden (1997).**

### 2.3 Model Inputs and Calculation Scenarios

**The nighttime atmospheric pressure, temperature, and densities of trace gases and main atmospheric constituents for calculation presented below were taken from the WACCM model (Marsh et al. (2013), Solomon et al. (2015)).**

The following sets of processes and rate coefficients were used in our calculations:

1. **(OH-N2 1Q) & (OH-O2 1Q) & R7: this model accounts for reactions R1, R2, R3, R6 and R7 from Table 1. R2 is treated as single-quantum ($v'$=1) process, R6 is also treated as single-quantum ($v' = v-1$). This model reproduces the initial model described by López-Puertas et al. (2004)**

2. **(OH-N2 3Q) & (OH-O2 1Q) & R7: same as model 1, however, R2 is treated as the three-quantum ($v'$=3) process and R6 is single-quantum ($v' = v - 1$). As it is shown below, this version matches best with the final model of López-Puertas et al. (2004) by increasing the efficiency factor to 3 for R2.**

3. **(OH-N2 3Q) & (OH-O2 MQ) & R7: same as model 2, however, R6 is treated as multi-quantum (any $v' \leq v - 1$ ) process.**

4. **(OH-N2 1Q) & (OH-O2 MQ) & R4,R5: reactions R1 through R6 from Table 1 are included. This is our basic model version with both "direct" (R2) and "indirect" (R4 + R5) mechanisms working together when R2 is treated as the single-quantum process ($v'$=1) as was suggested by (Kumer et al., 1978), however, R6 as the multi-quantum**

process (any $v' \leq v - 1$). **New mechanism R4 substitutes here the reaction R7 used in other models described above.**

5. **(OH-N2 3Q) & (OH-O2 MQ) & R4,R5: same as model 4, but "direct" process (R2) is treated as the three-quantum process corresponding to its 3 times higher efficiency suggested by López-Puertas et al. (2004).**

6. **(OH-N2 2Q) & (OH-O2 MQ) & R4,R5: same as model 5, however R2 is treated as the two-quantum process**

7. **(OH-N2 1,2Q) & (OH-O2 MQ) & R4,R5: same as model 5, however R2 is treated as two-quantum process for highly resonant transitions OH(9)+N$_2$(0)$\rightarrow$ OH(7)+N$_2$(2) and OH(2)+N$_2$(0)$\rightarrow$ OH(0)+N$_2$(2), and as single-quantum for all others.**

## 3 Results

### 3.1 Vibrational Temperatures of the CO$_2(\nu_3)$ levels

The vibrational temperature $T_\nu$ is defined from the Boltzmann formula

$$\frac{n_\nu}{n_0} = \frac{g_\nu}{g_0}\exp[\frac{E_\nu - E_0}{kT_\nu}],$$

which describes the excitation degree of level $\nu$ against the ground level 0. Here $g_\nu$ and $E_\nu$ are the statistical weight and the energy of level $\nu$, respectively. If $T_\nu = T_{kin}$ then level $\nu$ is in LTE.

Figure 1 shows the vibrational temperatures of the CO$_2$ levels of four isotopes, giving origin to 4.3 $\mu$m bands, which dominate the SABER nighttime signal (López-Puertas et al., 2004). These results were obtained for SABER scan 22, orbit 01264, 77°N, 03 March 2002. The same scan was used for the detailed analysis presented in the work by López-Puertas et al. (2004). The kinetic temperature retrieved for this scan from the SABER 15 $\mu$m radiances (SABER data version 2.0) and vibrational temperature of N$_2$(1) are also shown. **Solid lines in Fig. 1 represent simulations with our basic model [(OH-N2 1Q) & (OH-O2 MQ) & R4,R5], when both the "direct" process R2 (in its single-quantum version, as was suggested by Kumer et al. (1978)), and the new "indirect" process, reactions R4 + R5, are included. For comparison we also show vibrational temperatures (dashed lines) for the model (OH-N2 3Q) & ( OH-O2 1Q)] & R7 where the "indirect" mechanism is off and the "direct" process is treated as 3 quantum step, which is equivalent to the 3 times higher efficiency suggested by López-Puertas et al. (2004).**

Vibrational temperatures of CO$_2$ levels and N$_2$(1) depart from LTE around 65 km. **For both models vibrational temperatures nearly coincide up to 85-87 km, however, above this altitude, where the OH density is high, vibrational temperatures for [(OH-N2 3Q) & ( OH-O2 1Q)] & R7] are a few kelvin lower compared to those for [(OH-N2 1Q)& (OH-O2 MQ) & R4,R5]. These vibrational temperature differences explain differences of simulated CO$_2(\nu_3)$ emission for both models shown in Figure 2.** In both simulations, CO$_2$(00011) of main isotope 626 and N$_2$(1) have almost identical vibrational temperatures up to ~87 km which is caused by an efficient VV exchange (reaction (R3)).

## 3.2 The $CO_2$ 4.3 $\mu$m emission

Figure 2 displays our simulations of SABER channel 7 (4.3 $\mu$m) radiances for the WACCM model inputs which match the measurement conditions of the SABER scan described in Sect. 3.1. The calculations also account for the minor contribution in channel 7 radiation emitted by the OH($\nu \leq 9$) vibrational levels.

Our simulation for this scan with the [(OH-N2 1Q) & (OH-O2 1Q) & R7] set of rate coefficients is shown as the violet curve. The turquoise curve displays our results for the rate coefficient set [(OH-N2 3Q) & (OH-O2 1Q) & R7], which simulates the model suggested (López-Puertas et al., 2004) with the factor of 3 increased efficiency of reaction R2. One may see that treating R3 as a three-quantum VV process strongly enhances the pumping of the $CO_2(\nu_3)$ vibrations and the 4.3 $\mu$m radiance is in agreement with López-Puertas et al. (2004) results. These authors also reported that their results did not depend on whether R6 is treated as a single-quantum or, following Adler-Golden (1997), as a multi-quantum VV process. However, our calculations with the model [(OH-N2 3Q) & (OH-O2 MQ) & R7] do not confirm this feature: the blue curve, which displays this run, shows a significantly lower channel 7 signal. This is obviously caused by a much more efficient removal of the OH($v$) vibrational energy in the multi-quantum quenching by collisions with $O_2$. As a result, a significantly smaller part of this energy is collected by $N_2(1)$ and delivered to the $CO_2(\nu_3)$ vibrations with the "direct" mechanism R2. The higher channel 7 emission is, however, restored when we include the reactions R4 and R5 ("indirect" mechanism of energy transfer from OH($v$) to $CO_2(\nu_3)$) into the model and return R2 to its single-quantum mode. The red curve in Figure 2, which corresponds to our [(OH-N2 1Q) & (OH-O2 MQ) & R4,R5] model is nearly overlapped with the turquoise curve for the [(OH-N2 3Q) & (OH-O2 1Q) & R7] model. This demonstrates a very high efficiency of the "indirect" channel, compared to the "direct" one since it provides the same pumping of $CO_2(\nu_3)$ even when OH($v$) energy is efficiently removed in the multi-quantum version of R6.

We show in Figure 2 (black curve with diamonds), the channel 7 radiance profile for the SABER scan specified in Sec. 3.1. Comparing turquoise and red curves with this measurement, one may see that both the "direct" mechanism alone in its three-quantum version and the combination "indirect" and single-quantum "direct" mechanisms are close to the SABER radiance for this scan. However, to provide this pumping level, the multi-quantum "direct" mechanism needs to be supported by the inefficient single-quantum OH($v$) quenching in reaction R6 by collisions with $O_2$, which helps keeping higher population of OH(v). We also note here that both our violet [(OH-N2 1Q) & (OH-O2 1Q) & R7] and turquoise [(OH-N2 3Q) & (OH-O2 1Q) & R7] curves reproduce the corresponding results in Figure 10 of López-Puertas et al. (2004) (short-dash and solid lines, respectively) very well.

We also show in Figure 2 our study of how both the "direct" and "indirect" mechanisms work together when the "direct" process R2 is treated as multi-quantum. The magenta curve in this plot is the result obtained with the model [(OH-N2 3Q) & (OH-O2 MQ) & R4,R5] when R2 is treated as three-quantum process. This combination of both mechanism provides high $CO_2(\nu_3)$ pumping and subsequently strong Channel 7 emission. The latter exceeds the turquoise and read curves by 20-45% in the altitude range considered and strongly deviates from the measured radiance prifile.

Two other result of this study are shown only in the right panel of this plot for the signal differences. The light blue curve corresponds to simulations with the [(OH-N2 2Q) & (OH-O2 MQ) & R4,R5] model when the quantum transfer in R2 is reduced from 3 to 2. The dark green curve is obtained for the case when R2 is treated as two-quantum process for highly resonant transitions OH(9)+$N_2$(0)$\rightarrow$ OH(7)+$N_2$(2) and OH(2)+$N_2$(0)$\rightarrow$ OH(0)+$N_2$(2), and as single-quantum for all other vibrational levels. It is seen that both of these input versions bring the calculated radiance closer to our result for a single-quantum "direct" process R2 (red curve) and to our simulation [(OH-N2 3Q) & (OH-O2 1Q) & R7] of results obtained the López-Puertas et al. (2004) (turquoise).

In Figure 3 (upper and middle rows) we compare our simulation results for two sets of rate coefficients: (OH-N2 1Q) & (OH-O2 MQ) & R4 (red) and (OH-N2 3Q) & (OH-O2 1Q) (turquoise). The WACCM model nighttime inputs representing four different atmospheric situations described in Table 2 were used for these simulations. These inputs also match the measurement conditions of the four SABER nighttime scans (solar zenith angle > 105$^{o}$) listed in the Table 2. The corresponding SABER measured channel 7 4.3 $\mu$m radiances are shown in black as reference data.

One may see that in Figure 3 the "direct" mechanism alone with three-quantum efficiency for reaction R2, as well as both the "direct" (as single-quantum) and "indirect" mechanisms together provide similar results for all four atmospheric models, within a 10 to 30% difference range. By comparing these models to to measured radiances, both calculations are close to the observed signal down to 68 km for MLW and down to 75 km for SAW. For MLS and TROP the two-mechanism calculations are somewhat closer to measurements than those for "direct" mechanism alone in altitude interval 75-90 km.

### 3.3   Comparison of OH vibrational populations with ground and space-based observations

In Figure 4 we present relative OH($v$) populations calculated using three different sets of rate coefficients discussed in the previous section, which provided comparably high enhancement of the $CO_2(\nu_3)$ emission. These calculations are compared with the vibrational populations derived from ground (left-panel) and space-based (right panel) observations of OH emissions.

Measured populations (black) displayed in the left panel were recorded by Cosby and Slanger (2007) on 03 March 2000 using the echelle spectrograph and imager (ESI) on the Keck II telescope at Mauna Kea. The authors measured emission intensities of the 16 OH Meinel bands which were converted into the OH($v$) column densities and normalized to column density of OH($v$=9). Several observations of OH emissions were recorded throughout the night. We display the average column densities as well as their variation ranges for each vibrational level. The three simulated distributions (red, turquoise, and magenta) in this panel are modeled using WACCM inputs taken on 03 March 2000 at latitude 20°N at local midnight.

Measured densities displayed in the right panel of Figure 4 were taken from Migliorini et al. (2015) who analyzed VIRTIS (for Visible and Infrared Thermal Imaging Spectrometer) measurements on board the Rosetta mission. VIRTIS performed two limb scans of the OH Meinel bands from 87-105 km covering the latitude range from 38-47°N between 1:30 – 2:00 am, solar local time, in November 2009. To achieve high signal-to-noise, 300 radiance spectra (OH

$\Delta \nu = 1$ and 2) were collected and averaged. We show in the right panel of Figure 4 the OH($v$) population distribution normalized to OH($v$=9) derived in this study as well as corresponding uncertainties. The three simulated distributions (red, turquoise, and magenta) in this panel are modeled using WACCM inputs taken on 22 November 2000 at latitude 45°N at local midnight.

To simulate the ground-based observations of Cosby and Slanger (2007) (left panel) the calculated relative populations were integrated over the entire altitude range of our model (30-135 km). For for the right panel, we have integrated calculated OH($v$) densities from 87-105 km as observed by VIRTIS from Migliorini et al. (2015) to simulate mean population distribution obtained in this study.

The turquoise profiles in both panels of Figure 4 represent results obtained with our set of rate coefficients [(OH-N2
3Q) & (OH-O2 1Q) & R7] similar to the one used in López-Puertas et al. (2004), where the authors treated the OH-$N_2$ reaction with an efficiency increased by a factor of 3, the OH-$O_2$ reaction as single-quantum, and the OH-O reaction as a "sudden death" quenching or chemical OH removal process (reaction R7), with $v$ independent rate coefficient of $2.0 \times 10^{-10}$ cm$^3$/s. In the left panel, the turquoise profile shows higher relative populations compared to measurements for upper vibrational levels $\nu$>4, whereas in the right panel this model shows populations within the uncertainty range
of measurements for $\nu$>4. For lower vibrational levels $\nu \leq 4$, the populations calculated with this model are, however, significantly lower than measured ones: by 30% for $v$=3 in the left panel and by up to 85% for $v$=1 in the right panel. Significantly slower increase of populations calculated with the [(OH-N2 3Q) & (OH-O2 1Q) & R7] model compared to measurements can be explained by the lack of efficient mechanisms redistributing the OH($v$) energy from higher vibrations levels to lower ones. The single-quantum OH-$O_2$ reaction also allows for more excited OH molecules in the
upper vibrational levels relative to a multi-quantum process. Additionally, slower increase of calculated populations with the $v$ decreasing compared to measured ones which is seen in both panels, is the effect of the high quenching rate of coefficient of the reaction R7 for lower vibrational levels for which this reaction dominates over the single-quantum $O_2$ quenching.

The situation is different when our basic model (OH-N2 1Q) & (OH-O2 MQ) & R4,R5 is applied (red curves). As dis-
cussed above in the previous section, this model provides the same level of the $CO_2(\nu_3)$ emission pumping as the extreme model of (López-Puertas et al., 2004) (compare red and turquoise civws in Figure 2) however, demonstrate significantly different population distributions. Relative OH population distribution in the left panel shows our standard model in very good agreement with the results from Cosby and Slanger (2007), falling completely within the variation range of these measurements. The right panel also shows excellent agreement between calculations and measurements, where
former lie nearly completely within the measurement error bars for the majority of vibrational levels. In both panels our results reproduce well steady upward trend in populations from upper to lower vibrational levels. Significantly higher populations of lower OH levels in ths model is the result of redistribution of higher vibrational level energy to lower levels due to two dominant multi-quantum quenching mechanisms, namely the new reaction R4 and the multi-quantum version of reaction R6. We also note that R4 uses a lower rate coefficient than reaction R7 for quenching the
lower vibrational levels $\nu$<5 which results in maintaining their higher populations.

The population measured for $\nu=3$ (right panel) was the only level which showed disagreement with our model. Various reasons of increased measured population at $\nu=3$ are discussed by Migliorini et al. (2015), however, no definitive conclusions were given.

Above 90 km atomic oxygen density increases rapidly with the altitude. As a results the role of reaction R4 in quenching higher OH vibrational and pumping lower levels increases. This effect is easily seen in the right panel of Figure 4, where mean OH $(v)$ densities for higher altitude region 87-105 km are compared. The turquoise curve (no R4 reaction) in this panel shows lower populations compared to those calculated with with R4 included.

The magenta profiles in both panels represent our calculations with the model (OH-N2 3Q) & (OH-O2 MQ) & R4,R5 which is identical to our standard model except for the R2 reaction treated as the three-quantum one. The multi-quantum OH-N$_2$ VV transfer provides faster quenching of excited OH, hence, a lower overall population of the magenta profiles compared to red profiles. Despite showing reasonable agreement with measurements in both panels, this model caused, however, excessive increase for the 4.3 $\mu$m emissions, as seen in Figure 2.

### 3.4 OH 1.6 and 2.0 $\mu$m emissions

SABER channels 8 (2.0 $\mu$m), and 9 (1.6 $\mu$m) are dominated by the OH$(v)$ emission from levels $v$ = 8–9 and $v$ = 3–5, respectively. We simulated channel 8 and 9 radiances for four atmospheric models from Table 2. Results are shown in Figure 3, bottom row, as ratios of volume emission rates for channel 8 and 9. Volume emission rate (VER) is defined as the sum

$$R_{\mathbf{V}} = \sum A_{v,v'}[\mathbf{OH}(v)] = [\mathbf{OH}] \sum A_{v,v'} p_v$$

over all transitions contributing to the channel, where $p_v$= [OH$(v)$]/[OH] is the probability of the OH molecule to be in the vibrational state $v$. It follows from this expression that the VER ration does not depend on the total OH density and is, therefore, convenient for analyzing impacts of various populating/quenching mechanisms on OH(v) distribution. The calculations with our basic model [(OH-N2 1Q) & (OH-O2 MQ) & R4,R5] are shown in Figure 3 in red, and those for the model [(OH-N2 3Q) & (OH-O2 1Q) & R7] with the three-quantum mechanism R2 in turquoise, respectively. Black curves in this plot display SABER measured VER ratios, for which VERs were obtained with the Abel inversion procedure, similar to that described by López-Puertas et al. (2004), from the SABER channel 8 and 9 limb radiances for scans listed in Table 2.

Comparing red and turquoise profiles in Figure 3 (bottom row), one may see that our standard model (red) shows significantly lower VER ratios for altitudes 85-100 km than the model of López-Puertas et al. (2004) (turquoise). These differences between ratios are a result of very different OH$(v)$ population distributions (Figure 4) for each model, which were discussed in the previous section. The (Channel 8)/(Channel 9) VER ratios reflect these distributions very well since Channel 8 is sensitive to the OH$(v)$ emissions from higher levels 8 and 9, whereas Channel 9 records emissions from lower levels 3-5. Significantly higher population of lower vibrational levels in our model (red curves in Figure 4) explain low VER ratios. In contrast, the model [(OH-N2 3Q) & (OH-O2 1Q) & R7], which under-predicts lower level

populations, provides VER ratios which significantly exceed both our model results and measurements altitudes above 90 km, where [O] density rapidly increases with altitude. This comparison demonstrates the strong impact of reaction R2, which provides efficient quenching of higher OH vibrational levels in collisions with $O(^3P)$ atoms in this altitude region.

## 4    Conclusions

Kumer et al. (1978) first proposed the transfer of vibrational energy from chemically produced $OH(\nu)$ in the nighttime mesosphere to the $CO_2(\nu_3)$ vibration, $OH(\nu) \Rightarrow N_2(\nu) \Rightarrow CO_2(\nu_3)$. The effect of this "direct" mechanism on the SABER nighttime 4.3 $\mu$m emission was studied in detail by López-Puertas et al. (2004), who showed that in order to match observations , an additional enhancement is needed that would be equivalent to the production of 2.8-3 $N_2(1)$ molecules instead of one molecule for each quenching reaction $OH(\nu)+N_2(0)$. López-Puertas et al. (2004) concluded that the required 30% efficiency in the $OH(\nu)+N_2(0)$ energy transfer "...is, in principle, possible, although the mechanism(s) whereby the energy is transferred is (are) not currently known".

Recently, Sharma et al. (2015) suggested a new efficient "indirect" channel of the $OH(\nu)$ energy transfer to the $N_2(\nu)$ vibrations, $OH(\nu) \Rightarrow O(^1D) \Rightarrow N_2(\nu)$ and showed that it may provide an additional enhancement of the MLT nighttime 4.3 $\mu$m emission. Kalogerakis et al. (2016) provided a definitive laboratory confirmation of new $OH(\nu)$ + O vibrational relaxation pathway and measured its rate for $OH(\nu=9)+O$. **We included the new "indirect" energy transfer channel into our non-LTE model of the night time MLT emissions of $CO_2$ and OH molecules and studied in detail the impact of "direct" and "indirect" mechanisms on simulated vibrational level populations and radiances. The calculations were compared with (a) the SABER/TIMED nighttime 4.3 $\mu$m $CO_2$ and OH 1.6 and 2.0 $\mu$m limb radiances of MLT and (b) with the ground and space observations of the $OH(\nu)$ densities in the nighttime mesosphere. We found that new "indirect" channel provides significant enhancement of the 4.3 $\mu$m $CO_2$ emission, which is strong enough to fit SABER measured 4.3 $\mu$m radiances. This model also matches well with both SABER limb OH emission measurements and the ground and space observations of the $OH(v)$ densities in the mesosphere. Similarly strong enhancement of 4.3 $\mu$m emission can also be achieved with the "direct" mechanism alone assuming a factor of 3 increase in efficiency, as was suggested by López-Puertas et al. (2004). This model does not, however, reproduce either the SABER measured VER ratios of the OH 1.6 and 2.0 $\mu$m channels or the ground and space measurements of the $OH(v)$ densities. This mismatch is caused by the lack of efficient redistribution of the $OH(v)$ energy from the higher vibrational levels emitting at 2.0 $\mu$m to lower levels emitting at 1.6 $\mu$m in the models based on the "direct" mechanism alone. In contrast, this new "indirect" mechanism (reactions R4 and R5 of Table 1), efficiently removes at least 5 quanta in each $OH(v)+O(^3P)$ collision from high OH vibrational levels. Supported also by the multi-qauntum $OH(v)+O_2$ quenching (reaction R6 of Table 1), the new mechanism provides $OH(v)$ distributions which are in agreement with both measured VER ratios and observed $OH(v)$ populations.**

The results of our study suggest that the missing nighttime mechanism of $CO_2(\nu_3)$ pumping has finally been identified. This confidence is based on the fact that the new mechanism accounts for most of the discrepancies between measured and calculated 4.3 $\mu$m emission for various atmospheric situations, leaving relatively little room for other processes, among them the multi-quantum "direct" mechanism. The accounting for the multi-quantum transfer in re-action OH(v)+$N_2$ together with the "indirect" mechanism has little influence on the OH($v$) population distributions, however, can enhance the 4.3 $\mu$m emission. Therefore, further laboratory and/or theoretical investigation of this reaction is needed to define its role. Further improvements for the new "indirect" mechanism will require optimizing the set of rate coefficients used for OH($\nu$) relaxation by O($^3$P) and $O_2$ at mesospheric temperatures and, in particular, understanding the dependence of the "indirect" mechanism on the OH vibrational level. Relevant laboratory measurements and theoretical calculations are sorely needed to understand these relaxation rates and the quantitative details of the applicable mechanistic pathways. Nevertheless, the results presented here clearly demonstrate significant progress in understanding the mechanisms of the nighttime OH and $CO_2$ emission generation in MLT.

*Acknowledgements.* We would like to thank Ramesh Sharma for his helpful comments and productive collaboration with the new nighttime mechanism implementation. We would also like to thank Daniel Marsh for providing WACCM atmospheric inputs for our non-LTE model. The work by P.A.P was supported by the NASA grant NNX14AN71G and the NESSF Fellowship. The work by A.A.K. was supported by the NSF grant 1301762 and the NASA grant NNX15AN08G. The contributions of K.S.K were supported by NSF Grant 1441896.

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

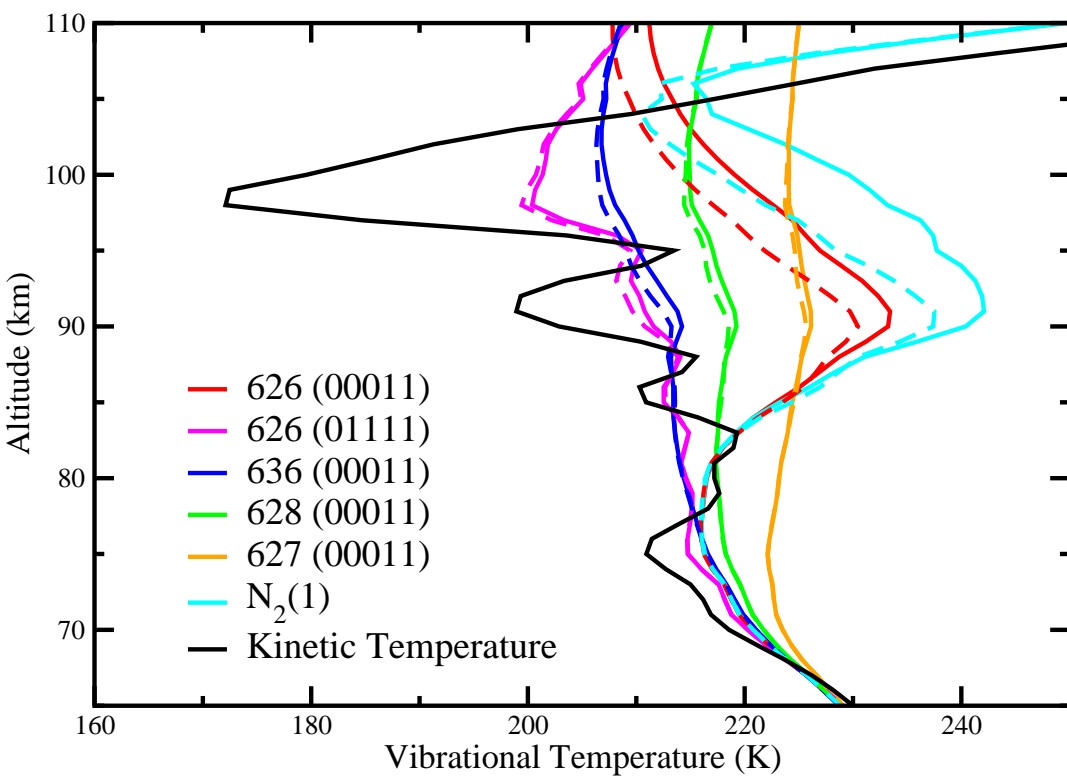

**Figure 1.** Nighttime vibrational temperatures of $CO_2(00011)$ of four $CO_2$ isotopes, $CO_2(01111)$ of main $CO_2$ isotope, and of $N_2(1)$ for SABER scan 22, orbit 01264, 77°N, 03 March 2002. Solid lines: [(OH-N2 1Q) & (OH-O2 MQ) & R4,R5]; dashed lines: [(OH-N2 3Q) & (OH-O2 1Q) & R7], see Section 2.3 for a description of calculation scenarios

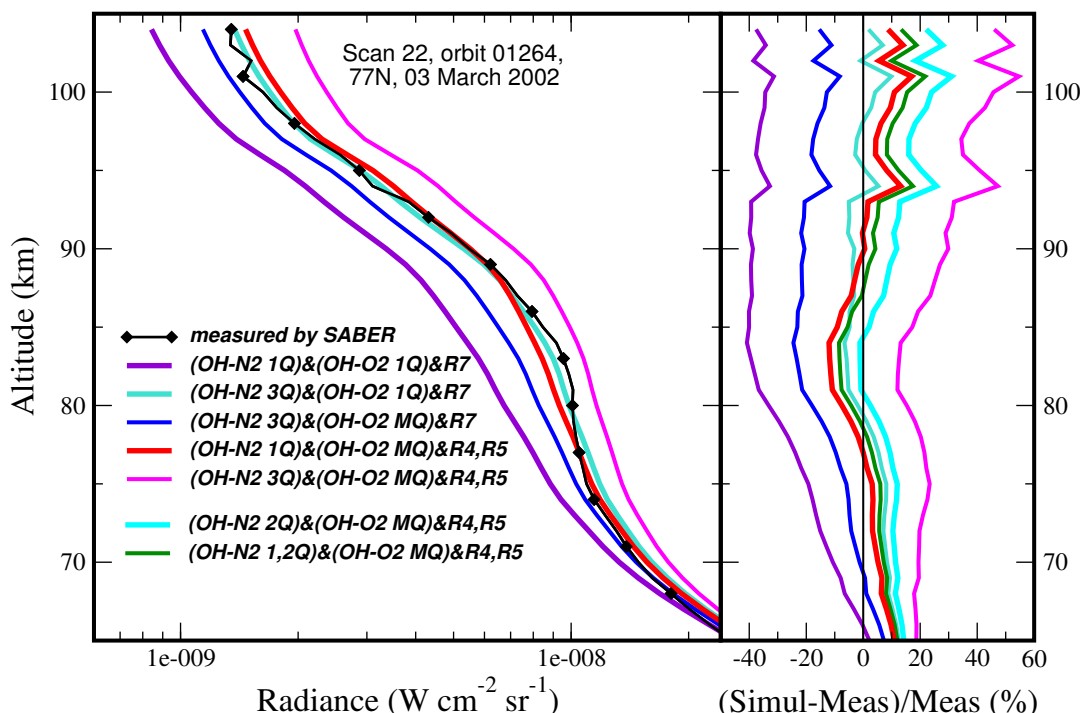

**Figure 2.** Left: measured and simulated SABER nighttime radiances in channel 7 (4.3 $\mu$m) for SABER scan 22, orbit 01264, 77°N, 03 March 2002. SABER measured (black); See Section 2.3 for a description of calculation scenarios displayed in the legend. Right: radiance relative difference (simulated-measured)/measured in percent.

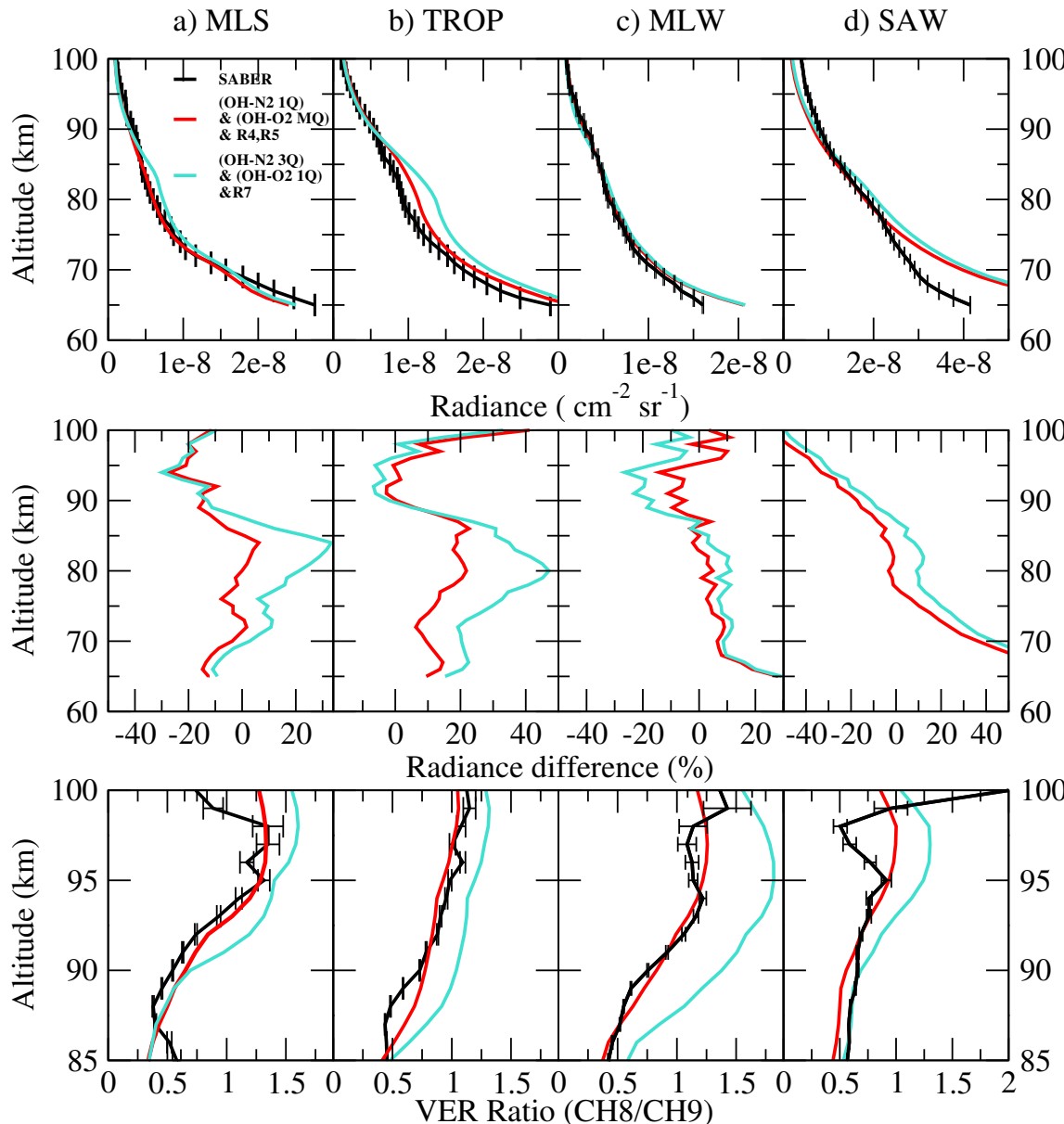

**Figure 3.** Top: measured and simulated nighttime radiances for SABER channel 7 (4.3 $\mu$m). Middle: radiance relative difference (simulated-measured)/measured in percent. Bottom: measured and simulated SABER Volume Emissions Rate Ratios (CH8/CH9). Four standard atmospheres are displayed: a) Mid-Latitude Summer (MLS), b) Tropical (TROP), c) Mid-Latitude Winter (MLW), and d) Sub-Arctic Winter (SAW) for selected SABER scans described in Table 2. SABER measured with NER (black); [(OH-N2 1Q) & (OH-O2 MQ) & R4,R5] (red); [(OH-N2 3Q) & (OH-O2 1Q) & R7] (turquoise); See Section 2.3 for a description of calculation scenarios

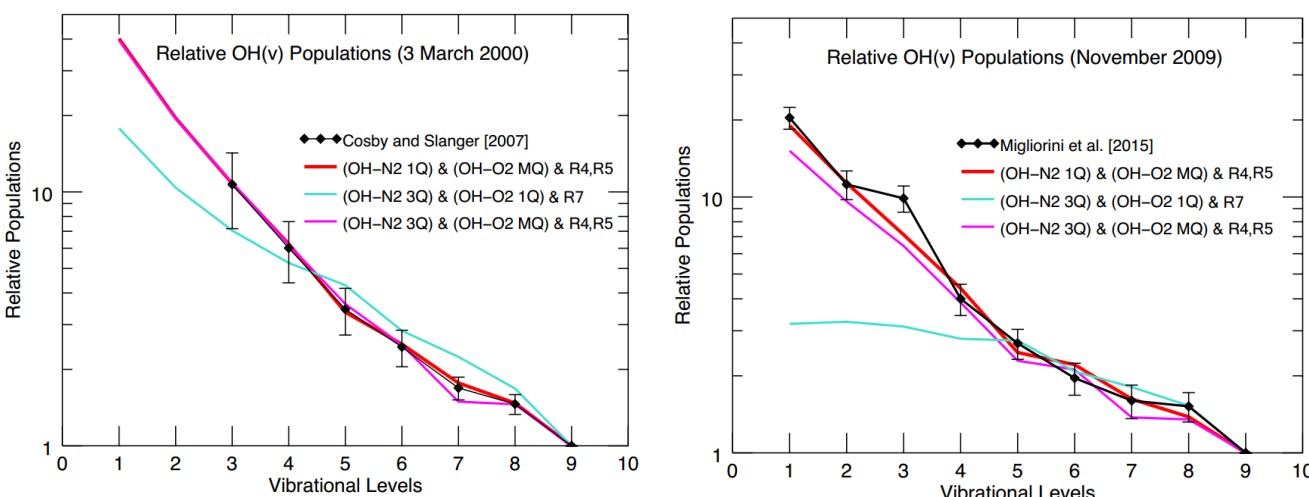

**Figure 4.** Relative OH populations, normalized to $\nu$=9, for measurements taken: (top) on 3 March 2000 at 20°N and (bottom) in November 2009 between 38-47°N. Measured populations (black with diamonds); Simulations: [(OH-N2 1Q) & (OH-O2 MQ) & R4,R5] (red); [(OH-N2 3Q) & (OH-O2 1Q) & R7] (turquoise); (OH-N2 3Q) & (OH-O2 MQ) & R4,R5] (magenta). See Section 2.3 for a description of calculation scenarios.

**Table 1.** Significant collisional processes used in model

| | Reaction | Reaction Rate ($cm^3sec^{-1}$) | Reference |
|---|---|---|---|
| (R1) | $H + O_3 \leftrightarrow OH(\nu \leq 9) + O_2$ | $k_1 = f_\nu^a \times 1.4 \times 10^{-10}\exp(-470/T)$ | Sander et al. (2011) & Adler-Golden (1997) |
| (R2) | $OH(\nu \leq 9) + N_2(0) \leftrightarrow OH(\nu - \nu') + N_2(\nu')$ $\nu'=1,2,3$ | $k_2 = f_\nu^b \times 1.4 \times 10^{-13}$ | Adler-Golden (1997) & Lacoursière et al. (2003) |
| (R3) | $N_2(1) + CO_2(0) \leftrightarrow N_2(0) + CO_2(\nu_3)$ | $k_3 = 8.91 \times 10^{-12} \times T^{-1}$ | Shved et al. (1998) |
| (R4) | $OH(\nu \geq 5) + O(^3P) \leftrightarrow OH(0 \leq \nu' \leq \nu\text{-}5) + O(^1D)$ | $k_4 = f_\nu^c \times (2.3 \pm 1) \times 10^{-10}$ | Kalogerakis et al. (2016) & Sharma et al. (2015) |
| | $OH(\nu < 5) + O(^3P) \leftrightarrow OH(0) + O(^3P)$ | $k_4 = 5.0 \times 10^{-11}$ | Caridade et al. (2013) |
| (R5) | $O(^1D) + N_2(0) \leftrightarrow O(^3P) + N_2(\nu)$ | $k_5 = 2.15 \times 10^{-11}\exp(110/T)$ | Sander et al. (2011) |
| (R6) | $OH(\nu \leq 9) + O_2(0) \leftrightarrow OH(\nu') + O_2(1)$ $\nu'=0,1,2, ... \nu\text{-}1$ | $k_6 = f_\nu^d \times 1.18 \times 10^{-13}$ | Adler-Golden (1997) |
| (R7) | $OH(\nu \leq 9) + O(^3P) \leftrightarrow OH(0) + O(^3P)$ | $k_7 = 2.0 \times 10^{-10}$ | Adler-Golden (1997) |

[a] $f_\nu(\nu=5\text{-}9) = (0.01, 0.03, 0.15, 0.34, 0.47)$

[b] $f_\nu(\nu=1\text{-}9) = (0.06, 0.10, 0.17, 0.30, 0.52, 0.91, 1.6, 7, 4.8)$

[c] $f_\nu(\nu=5\text{-}9) = (0.91, 0.61, 0.74, 0.87, 1.0)$

[d] $f_\nu(\nu=1\text{-}9) = (1.9, 4, 7.7, 13, 25, 43, 102, 119, 309)$

**Table 2.** Selected nighttime SABER scans

| | Atmosphere | Latitude | Day | Orbit | Scan |
|---|---|---|---|---|---|
| (a) | Mid-Latitude Summer (MLS) | 37°S | 26 Jan 2004 | 11556 | 62 |
| (b) | Tropical (TROP) | 6°N | 20 Jan 2008 | 33130 | 48 |
| (c) | Mid-Latitude Winter (MLW) | 34°S | 15 Jul 2010 | 46594 | 90 |
| (d) | Sub-Arctic Winter (SAW) | 72°S | 15 Jul 2010 | 46585 | 78 |