# Peer review of "Resolving the mesospheric nighttime 4.3 $\mu$ m emission puzzle: Comparison of the CO2( $\nu_3$ ) and OH( $\nu$ ) emission models"

_Atmospheric Chemistry and Physics, 2016_

## Referee Comment (RC1) · Anonymous Referee #2 · 19 Dec 2016

This study investigates the impact of the additional vibrational energy transfer channel OH* → O($^1$D) → N$_2$(v=1) → CO$_2$(v$_3$), as recently proposed by theroretical and laboratory studies, on CO$_2$ 4.3 $\mu$m nightime radiances by means of NLTE radiative transfer simulations in comparison with SABER Channel 7 measurements. The authors show that the inclusion of the proposed new mechanism improves noticably the agreement between simulated and observed radiances.

The manuscript is generally well written and the topic is of high interest because of its implication for the analysis of measured 4.3 $\mu$m radiances (and potentially for the retrieval of CO$_2$ nighttime densities in the MLT). However, I have one major comment which needs to be addressed: The authors state on p3, line 27 (and also p5, line 26)

that OH is taken from WACCM results. However, what is required here is OH$^*$(v=1-9), likely not available from WACCM simulations. Further, the text in Section 2.2 suggests that OH$^*$(v=1-9) populations have been calculated based on kinetic rates provided in Table 1. This means that the relevant input for such calculations is H, $O_3$, and O (the latter taken from SABER, as stated in Section 2.1). Have the required H and $O_3$ profiles been taken from WACCM? If this was the case, it might be possible that OH$^*$ excitation is underestimated since WACCM $O_3$ was shown to be in tendency lower (by about a factor of 2) than observations (Smith et al., 2014). Given the importance of the actual amount of excited OH on the simulated 4.3 $\mu$ radiances, this point needs to be clarified.

Further, since SABER provides also independent measurements of OH$^*$ (Channel 8 and 9), a direct validation of the calculated OH$^*$ densities is feasible (as done in the López-Puertas et al., 2004 study) and should be undertaken.

Minor comments:

Table 1, footnote b: There seems to be a typo in f$_v$ for v=8. According to Adler-Golden (1997) this factor should read 2.7 (instead of 7)

Smith, A. K., M. López-Puertas, B. Funke, M. García-Comas, M. G. Mlynczak, and L. A. Holt (2015), Nighttime ozone variability in the high latitude winter mesosphere, J. Geophys. Res. Atmos., 119, 13,547–13,564, doi:10.1002/ 2014JD021987.
* * *

---

## Referee Comment (RC2) · Anonymous Referee #3 · 20 Dec 2016

General comment.

The manuscript proposes an alternative mechanism to that of Lopez-Puertas et al. (2004) (LP04) for explaining the $N_2(v=1)$ excitation that gives rise to an enhanced $CO_2$ 4.3 $\mu$m night-time emission as measured by SABER. Such mechanism is compatible with that proposed by those authors but the energy is transferred through an intermediate pathway. It then represents an important research finding that deserves to be published.

I do not fully agree however in the way it is presented at some passages. It gives the impression that the presented mechanism is the "correct" one and the previously proposed mechanism is not correct. So far only some theoretical estimates have been

carried out suggesting that the multi-quantum energy transfer from OH(v) to N2(1) is not likely, but no laboratory measurements have corroborated it. I would then be not so categorical about the new mechanism with sentences such as "... the missing night-time mechanism of CO2(v3) pumping has finally been identified."

I think the paper should be presented as being able to explain the SABER radiances with a plausible mechanism for indirect transfer of the energy from OH(v) to N2(1) instead of the direct multi-quantum energy transfer with a required efficiency of 2.8-3 as suggested by LP04.

The authors should also be cautious with assertions such as the new mechanism "improves agreement with SABER observations (in the title, as well as in the conclusions)". Both mechanisms seem to be able to explain SABER radiances with a very similar degree of agreement (Fig. 2 of the manuscript and Fig. 12 of LP04). It seems the new pathway is more plausible according to some theoretical estimates but I have not seen in the manuscript a clear discussion about why the multi-quantum mechanism should be ruled out.

Furthermore, it should be proved more quantitatively that the new proposed mechanism, that affect to the OH(v) populations, is able to explain the SABER OH measurements and that it is consistent with the multiple previous rocket measurements of OH(v).

I think these points should be addressed before the manuscript is accepted for publication.

Specific comments.

1) I think the title should be revised. The new model calculations are equally good as previous ones in reproducing SABER 4.3 $\mu$m observations. The focus should be put on the new OH(v) => N2(1) transfer mechanism rather than on the reproduction of the radiances. The agreement of the new calculations with SABER (-20,30%) are

not better that the results shown by LP04 in their Fig. 12, that in spite of the larger uncertainties in the "theoretical" reaction rates for OH(v) to O(1D) energy transfer as well as in the O abundance.

Also, I do not find appropriate the other part of the title: "Resolving the mesospheric night-time 4.3 $\mu$m emission puzzle". Where is the puzzle? LP04 already explained SABER radiances to within +/-20%. I would be more in favour of a title like "A new alternative mechanism for explaining the mesospheric night-time 4.3 $\mu$m emission" or even been being more precise, for explaining the mesospheric night-time excitation of N2(1)".

Abstract. Although the transfer of energy mentioned in the manuscript of OH(v)=> N2(v)=>CO2(v3)=4.3 $\mu$m emission, is correct it could be simplified to OH(v)=> N2(v), since the mechanism proposed affects only to this part of the transference and the remaining transfers, =>CO3(v3) => 4.3 $\mu$m, are common with the previous study.

Lines 6-8. "A previous study suggested the "direct" transfer OH(v) => N2(v) => CO2(v3) => 4.3 $\mu$m of vibrational excitation from OH(v) to CO2 in the night-time mesosphere. However, accounting for this excitation mechanism alone leads to significant under-prediction (by up to 80%) of observed 4.3 $\mu$m limb radiances." See comment above. If the same mechanism is assumed as multi-quantum (with an efficiency of 2.8-3) LP04 were able to explain the SABER radiances. Hence, that sentence should be re-written. That mechanism with single quantum was not the final conclusion of LP04. Somehow the manuscript is inconsistent as this assertion is correctly mentioned in other parts or the text but not everywhere, as in theses sentences, and other important instances, as in Fig. 2.

Lines 13-14. "This finding creates new opportunities for the application of CO2 4.3 $\mu$m observations in the study of the energetics and dynamics of the night-time MLT." I am not really convinced about that. Even if the energy pathway from OH(v) to N2(1) was not clear, the previous mechanism was already able to explain the measured SABER

4.3 $\mu$m night-time radiances as well as with the new alternative mechanism.

Page 2. Par. from lines 3 to 13. I would not argue as a motivation for this research its potential use for retrieving CO2 from night-time 4.3 $\mu$m SABER measurements. I think the new mechanism is already important on its own, i.e., it is important to understand the non-LTE processes occurring in the middle atmosphere, and it does not need the motivation of CO2 night-time retrieval because this presents additional problems, which, in my view, are more important. First, I think to measure CO2 at night-time is not very important as far as we have daytime measurements for the wanted latitudes/seasons. Because, as it is very well mentioned in the manuscript, CO2 has a very long chemical lifetime, we do not expect significant (photochemical) diurnal variations. Only tides, but they would also be present in night-time observations. The only region of interest would be the polar winter, where no daytime measurements are available yet. But the retrieval of CO2 there has other problems. As it is in high latitudes, auroral excitation of N2(1) is very important and that is not well known. Also the geomagnetic conditions are very variable and hence difficult to model. In addition, as has been demonstrated by Winick et al. (1988), the location of the aurora along the LOS has to be known very well. Furthermore, most of the night-time 4.3 $\mu$m radiance comes from the strong CO2(v3) fundamental band, which is very optically thick. And last, the night-time 4.3$\mu$m signal is usually much more noisy ($\sim$a factor for 100 or larger) than the daytime one.

Page 2. Line 26 and ff. "However, using laboratory rate coefficients of corresponding reactions the authors were unable to reproduce the 4.3 $\mu$m radiance observed by SABER." This is only partially correct, as they were able to reproduce SABER radiances when using an efficiency of 2.8-3 with the same reaction rate.

Reaction R2 has been normally used as OH(v,<=10) + N2(0) <=> OH(v-1) + N2 note that N2 is not excited, see, e.g. Adler-Golden et al (1997), because it has been used in OH(v) modelling and the interest was the deactivation of OH(v), without paying any attention to the final state of N2, e.g., if it was excited or not excited. Hence, I think the

statement (line 30) that "its has been accepted with a value of 1" needs more discussion. It has been used most of the times regardless of the excitation of N2. Theoretical estimates by Adler-Golden et al. (1997) and Sharma et al. (2015) suggest that it takes place at single-quantum relaxation. However, to my knowledge, the efficiency of this reaction has not been measured in the laboratory, mainly because the major interest was to know the relaxation of OH(v) and not where the energy goes. Are these reasons enough for completely disregarding the multi-quantum? I do not think so. The mechanism the authors propose sounds plausible but one should be careful about assuring that it is "the" mechanism (and reject the LP04 mechanism). If still the authors would like to be categorical, I think this point needs to be discussed deeper in the manuscript.

Page 3.

Minor comment. Lines 1-2. The proposed new mechanism strictly refers to OH(v) to N2(1), rather than OH(v) to CO2(v3).

Lines 5-6. "Kalogerakis et al. (2016) provided a definitive laboratory confirmation for the validity of this new mechanism." Were they able to measure the reaction rate and energy efficiency of this mechanism? Is this new mechanism still based on the "theoretical" calculations of Sharma et al. (2015) for the reaction rate of the $OH(v)+O(3P) => OH(v')+O(1D)$ ?

Lines 7-8. If the author would like to be consistent with the model of LP04 they should use the efficiency of 2.8-3.

Line 8. "... OH(v) energy transfer to "N2(1)" instead of to "CO2".

Lines 19. Again, in order to be consistent with LP04 the authors should use an efficiency of 2.8-3.

Lines 25-27. The SABER data version should be stated.

Line 27. As the authors mentioned above CO2 has been retrieved. I was then expecting to use the retrieved CO2 instead of that of WACCM. This should give better

simulated radiances and remove some uncertainties.

Reactions R1-R4 are repeated in the text and in the Table. Maybe they should be kept only in the Table.

Page 4.

Sec. 2.2

A major comment. As the new proposed mechanism affects also to the population of OH(v) and the emissions from these levels were measured by SABER in two different channels, I think it is essential that the authors demonstrate that the new OH(v) model explain very well the measured SABER OH radiances, as LP04 did. Thus, figures should be presented for different conditions comparing SABER observations and modelled radiances for the two OH SABER channels.

About the O(3P) abundance and the OH(v) model, the authors state that they used the O(3P) retrieved from SABER measurements. The SABER O(3P) is derived from the SABER OH radiances but a photochemical OH(v) model is required for such inversion (Mlynczak et al., 2013). How do the reaction rates for the OH(v) model used here (Table 1) compare to those of Mlynczak et al., 2013? Actually, to be consistent, it should be used the same photochemical OH(v) model in both cases.

Along this line, the mechanism proposed by LP04 did not affect the established OH(v) model (e.g. Adler-Golden et al., 1997), so in that sense it was also compatible with most of previous OH(v) emission rocket measurements. How does the new OH(v) photochemical model compare to that of Adler-Golden et al.? I.e., it is also compatible with previous OH(v) emission rocket measurements?

Line 9. The text in this line is repeated a few lines below.

Lines 14-15. Could the authors be more precise with "lower" and "higher" CO2 vibrational levels?

Line 21. Then the values used for the reaction rate of the new mechanism are based on theoretical estimations? not measured values? Kalogerakis et al. (2016) did not measure those the reaction rates and efficiencies? If measured, why not use the measurements with their errors instead of theoretical estimates?

Page 5. Sec. 3.2 Lines 20-25. This section and Fig. 2, when the authors refer to the calculations of LP04 with the "direct" mechanism, can be misleading. LP04 were able to reproduce the observed SABER radiances when using this mechanism but with an efficiency of 2.8.

"... inputs identical to those of Lopez-Puertas et al (2004)." Lopez-Puertas et al (2004) used version 1.03 of SABER parameters. Which version has been used here? Are they really identical? To which degree?

Lines 25 and ff. Using OH densities from WACCM. I believe the authors mean OH(v) densities, i.e. vibrationally excited OH, not OH in the ground state. In this case, the WACCM OH(v) densities might be largely under-predicted since it is well known that WACCM mesospheric $O_3$ abundance is underestimated by at least a factor of 2 with respect to satellite measurements (both SABER and MIPAS) (Smith et al., 2013). By the way, the authors describe the OH(v) photochemical and the sources of some atmospheric constituents but not the source of $O_3$ and H. Or was it included OH(v) (instead of OH) from WACCM and the OH photochemical model described (Table 1) is that of WACCM?

Page 7. Conclusions.

Lines 16-17: "This significant improvement suggests that the missing night-time mechanism of $CO_2(v_3)$ pumping has finally been identified. " I would not be so categorical. At least experimentally it has not yet been ruled out the possibility of multi-quantum energy transfer from OH(v) to $N_2(1)$.

"Relevant laboratory measurements and theoretical calculations are sorely needed to

understand these relaxation rates and the quantitative details of the applicable mechanistic pathways." I understood from this manuscript that these have been already done (e.g. Sharma et al., 2015 and Kalogerakis et al. (2016). What new is needed?

"Nevertheless, results presented here clearly demonstrate significant progress in understanding the generation mechanisms of the night-time $CO_2$ 4.3 $\mu$m emission and represent an important step towards developing the algorithm(s) suitable for retrieving $CO_2$ densities in the MLT from the SABER night-time limb radiances." I agree with the first part of the sentence. However, I see no real progress for an eventual retrieval of $CO_2$ from night-time radiances. SABER measurements were already reproduced before as good as with this new mechanism. Further, even with such a good reproduction, the inversion of $CO_2$ from night-time 4.3 $\mu$m emission in the regions where it would be useful (winter polar night) is still very difficult due to the reasons mentioned above.

References Smith, A. K., Harvey, V. L., Mlynczak, M. G., Funke, B., Garcia-Comas, M., Hervig, M., Kaufmann, M., Kyrola, E., Lopez-Puertas, M., McDade, I., Randall, C. E., Russell, J. M., III, Sheese, P. E., Shiotani, M., Skinner, W. R., Suzuki, M. and Walker, K. A.: Satellite observations of ozone in the upper mesosphere, J. Geophys. Res., 118(11), 5803'Åì5821, doi:10.1002/jgrd.50445, 2013.

Winick, J.R., Picard, R.H., Sharma, R.D. et al. (1988), Radiative transfer effects on aurora enhanced 4.3 microns emission, Adv. Space Res. 7, (10)17.

---

## Author Comment (AC2) · 10 Mar 2017

We thank the reviewer for careful reading of the manuscript and helpful suggestions that lead to an improvement of the text. Here we reproduce referee's comments in full in italic and show our replies. Similarly, in the manuscript we use bold font to clearly denote the changed text.

*I have one major comment which needs to be addressed: The authors state on p3, line 27 (and also p5, line 26) that OH is taken from WACCM results. However, what is required here is OH∗(v=1-9), likely not available from WACCM simulations. Further, the text in Section 2.2 suggests that OH∗(v=1-9) populations have been calculated based on kinetic rates provided in Table 1. This means that the relevant input for such calculations is H, O3 , and O (the latter taken from SABER, as stated in Section 2.1). Have the required H and O3 profiles been taken from WACCM? If this was the case, it might be possible that OH∗ excitation is underestimated since WACCM O3 was shown to be in tendency lower (by about a factor of 2) than observations (Smith et al., 2014). Given the importance of the actual amount of excited OH on the simulated 4.3 µ radiances, this point needs to be clarified.*

We have really missed to specify in the text where from H and O3 was taken for our calculations. We now added to the paper text that both H and O3 were taken from SABER retrievals.

*Further, since SABER provides also independent measurements of OH∗ (Channel 8 and 9), a direct validation of the calculated OH∗ densities is feasible (as done in the López-Puertas et al., 2004 study) and should be undertaken.*

We have addressed this comment in our replies to the report of referee 2. We reproduce it here.

The referee is raising a good point that we considered in our consistency checks by evaluating the effect on the OH(v) population for the two highest vibrational levels. We found very good agreement (within a few percent) between the number density determined for OH(8+9) from our test data sets and the SABER channel 8 observations. The absolute number density for OH(8+9) can be directly determined from the SABER Channel 8 radiances (Mast et al., 2013) and is therefore a rigorous consistency check. The extracted OH(8+9) absolute number density does not depend on any previous model result (only the radiance profile and the almost equal Einstein emission coefficients A97 and A86 are needed). Our ultimate goal is to develop an updated model that handles simultaneously all the SABER OH(v) and CO2 emission channels, but this is clearly out of the scope of this report. The goal of our study was to estimate, with the help of model calculations that are based on reliable inputs, the effect of the recently discovered "indirect" pumping mechanism of N2(v) at nighttime. The latter was suggested and then experimentally confirmed by Sharma et al, 2015 and Kalogerakis et al, 2016, respectively. We believe we demonstrated the importance of this new mechanism and made significant contributions that complement the previous studies. Nevertheless, as we make clear in our conclusion statements, there is plenty of room for further research.

*Minor comments:*

*Table 1, footnote b: There seems to be a typo in fv for v=8. According to Adler-Golden (1997) this factor should read 2.7 (instead of 7)*

The value 2.7 appears only once  in the paper *Adler-Golden (1997),* namely in the Table 1, column 2, for the reaction OH(v=2)+O2. The footnote b in our table refers, however, to the values taken from

column 3 of same Table 1 of *Adler-Golden (1997)* for the reaction $OH(v)+N_2$, where the value 7 corresponds to v=8.

---

## Author Comment (AC3) · 10 Mar 2017

We thank the reviewer for careful reading of the manuscript and helpful suggestions that lead to an improvement of the text. Here we reproduce referee's comments in full in italic and show our replies. Similarly, in the manuscript we use bold font to clearly denote the changed text.

**General comment**

*The manuscript proposes an alternative mechanism to that of Lopez-Puertas et al. (2004) (LP04) for explaining the N2(v=1) excitation that gives rise to an enhanced CO2 4.3 μm night-time emission as measured by SABER. Such mechanism is compatible with that proposed by those authors but the energy is transferred through an intermediate pathway. It then represents an important research finding that deserves to be published.*
*I do not fully agree however in the way it is presented at some passages. It gives the impression that the presented mechanism is the "correct" one and the previously proposed mechanism is not correct. So far only some theoretical estimates have been carried out suggesting that the multi-quantum energy transfer from OH(v) to N2(1) is not likely, but no laboratory measurements have corroborated it. I would then be not so categorical about the new mechanism with sentences such as "... the missing night-time mechanism of CO2(v3) pumping has finally been identified."*
*I think the paper should be presented as being able to explain the SABER radiances with a plausible mechanism for indirect transfer of the energy from OH(v) to N2(1) instead of the direct multi-quantum energy transfer with a required efficiency of 2.8-3 as suggested by LP04.*
*The authors should also be cautious with assertions such as the new mechanism "improves agreement with SABER observations (in the title, as well as in the conclusions)". Both mechanisms seem to be able to explain SABER radiances with a very similar degree of agreement (Fig. 2 of the manuscript and Fig. 12 of LP04). It seems the new pathway is more plausible according to some theoretical estimates but I have not seen in the manuscript a clear discussion about why the multi-quantum mechanism should be ruled out.*
*Furthermore, it should be proved more quantitatively that the new proposed mechanism, that affect to the OH(v) populations, is able to explain the SABER OH measurements and that it is consistent with the multiple previous rocket measurements of OH(v).*
*I think these points should be addressed before the manuscript is accepted for publication.*

**Response to general comment**

There appears to be some confusion regarding the manuscript of Lopez-Puertas et al. (LP04) and its findings. The referee keeps making references to an unspecified multi-quantum alternative mechanism of LP04. Because of the plethora of "alternative facts" these days, it seems most appropriate to revisit the statements of LP04 (direct quotations from LP04 in boldface italics).

This is how LP04 summarize the findings or that study in the abstract:

*"…We have investigated the SABER 4.3 μm radiances with the help of a non-LTE radiative transfer model for $CO_2$ and found that the large radiances can be explained by a fast and efficient energy transfer rate from OH(v) to $N_2$(1) to $CO_2(v_3)$, whereby, on average, 2.8–3 $N_2$(1) vibrational quanta are excited after quenching of one OH(v) molecule. A series of alternative excitation mechanisms that may enhance the nighttime 4.3 ▯m limb radiance were considered and found to be insignificant. The mechanism(s) whereby the energy is transferred from OH(v) to $N_2$(v) is (are) still uncertain…"*

And below is an excerpt from the conclusions of LP04:

*"…The indirect contribution of OH(v) through vibrational relaxation to $N_2$ and subsequent transfer to $CO_2$(v3) significantly enhances the $CO_2$ 4.3 mm limb radiance in and everywhere above the upper mesosphere. However, the energy transfer estimated from the currently accepted quenching rates of OH(▯) by $N_2$ is not enough to explain the large SABER radiances. An energy transfer from OH(v) to $N_2$ that is more efficient than currently assumed, whereby a single $N_2$(1) molecule is excited after the relaxation of any OH(v) level, is required to explain the 4.3 μm radiance. On average, about 2.8 − 3 $N_2$(1) molecules per OH(v) molecule are required to explain the SABER radiances.*

There is no alternative mechanism presented in any detail in LP04. The important contribution of the LP04 study is that it examined in detail all the excitation mechanisms known at that time and, after de-termining that all these were inadequate, the LP04 study quantified the energy transfer rate that would be required to account for the discrepancies between model calculations and observations. Neverthe-less, no detailed mechanism was described in LP04, let alone validated or justified by referring to any relevant theoretical or experimental investigations. As the quotes above show clearly, LP04 concluded that the nature of the active energy transfer mechanism was not known at that time. We suggest the ref-eree read more carefully the LP04 paper so as not to misrepresent the important contributions of that study.

We agree with the referee that this manuscript "*represents an important research finding that deserves to be published.*"

**Specific comments**

*I think the title should be revised. The new model calculations are equally good as previous ones in reproducing SABER 4.3 μm observations. The focus should be put on the new OH(v) => N2(1) transfer mechanism rather than on the reproduction of the radiances.*

Our title does not state that we are"*reproducing SABER 4.3 μm observations*"-.  We only say that "new model calculations improve agreement  with SABER observations".

*The agreement of the new calculations with SABER (-20,30%) are not better that the results shown by LP04 in their Fig. 12, that in spite of the larger uncertainties in the "theoretical" reaction rates for OH(v) to O(1D) energy transfer as well as in the O abundance.*

The reviewer obviously means here that "*the results shown by LP04 in their Fig. 12*" obtained for some selected scans using the efficiency factor f=3 for the N2(1) production in the "direct" mechanism. *LP04* showed that using this factor as a fitting parameter removes about 40% of differences for these particular scans. We, however, showed in this study that for various latitudes/seasons the  differences

between measured and modeled radiances, when f=1 is applied, reach up to 80%. These large differences can be reduced by using values of f=6 and higher. This kind of arbitrary fitting, however, is not needed anymore, when the new "indirect" mechanism is added to the standard one by Kumer (with f=1): working together both mechanisms provide reliable agreement for the large variety of atmospheric conditions. **This reviewer obviously does not see this qualitatively new situation** and persistently returns to this point many times in his/her report.

*Also, I do not find appropriate the other part of the title: "Resolving the mesospheric*
*night-time 4.3 µm emission puzzle". Where is the puzzle?*
*LP04 already explained SABER radiances to within +/-20%.*

LP04 precisely formulated this puzzle: "***An energy transfer from OH(v) to $N_2$ that is more***
***efficient than currently assumed is required to explain the 4.3 µm radiance"***.
It actually took 11 years before a new efficient mechanism was found. Was it not a really long lasting puzzle? (even this statement neglects the fact that the puzzle started with Kumer as far back as the 1970s).

*I would be more in favour of a title like "A new*
*alternative mechanism for explaining the mesospheric night-time 4.3 µm emission" …*

An "alternative mechanism" means that the problem can be explained by one or another way.
Following persistent return of the referee to this point we will show below that the
new mechanism leaves little room for alternative explanations. Does the reviewer suggest that after a new mechanism has been identified and validated, we simply forget about it? We have nothing against alternative explanations that will be shown to improve upon the new efficient energy transfer.

*...or even been being more precise, for explaining the mesospheric night-time excitation of*
*N2(1)".*

Most of the authors have worked for many years with the analysis of SABER measurements (trying to resolving this and other "puzzles" related to these observations). We believe the title, which states improved agreement between SABER measurements and modeling is more accurate than the more academic "*night-time excitation of N2(1)*". Therefore, we prefer to keep our current tittle, however, added in the conclusion words about the progress in "*explaining the mesospheric night-time excitation of N2(1)*".

*Abstract. Although the transfer of energy mentioned in the manuscript of OH(v)=>*
*N2(v)=>CO2(v3)=4.3 µm emission, is correct it could be simplified to OH(v)=> N2(v),*
*since the mechanism proposed affects only to this part of the transference and the*
*remaining transfers, =>CO3(v3) => 4.3 µm, are common with the previous study.*

Following this advice, we changed the text in the paper body. However, in the abstract we simply want to let the reader know that we study the transfer of the OH(v) vibrational energy to CO2 and further to 4.3 µm emission. We, therefore, prefer to keep here the reaction chain as it is. Meanwhile, Kumer at al 1978 put the same reaction chain even in that paper title.

*Lines 6-8. "A previous study suggested the "direct" transfer OH(v) => N2(v) => CO2(v3)*
*=> 4.3 µm of vibrational excitation from OH(v) to CO2 in the night-time mesosphere.*

*However, accounting for this excitation mechanism alone leads to significant under-
prediction (by up to 80%) of observed 4.3 μm limb radiances."*

Following this comment, we changed the text of the  abstract. We write now "However, accounting for this excitation mechanism (**with the currently accepted efficiency**) alone leads ..."

*If the same mechanism is assumed as multi-quantum (with an efficiency of 2.8-3) LP04
were able to explain the SABER radiances. Hence, that sentence should be re-written.*

Please see our discussion below about possible *multi-quantum* nature of these mechanisms. Relying on it we do not see the reasons to re-write this sentence.

*That mechanism with single quantum was not the final conclusion of LP04. Somehow
the manuscript is inconsistent as this assertion is correctly mentioned in other parts or
the text but not everywhere, as in theses sentences, and other important instances, as
in Fig. 2.*

Above, we reproduced the LP04 conclusions. The final conclusion of that previous study was emphasized as a need for a more efficient mechanism, which we believe has finally been found. Following this comment we tried to make the paper text more consistent.

*Lines 13-14. "This finding creates new opportunities for the application of CO2 4.3 μm
observations in the study of the energetics and dynamics of the night-time MLT." I am
not really convinced about that. Even if the energy pathway from OH(v) to N2(1) was
not clear, the previous mechanism was already able to explain the measured SABER
4.3 μm night-time radiances as well as with the new alternative mechanism.*

The same statements about an "altenative mechanism" is repeated here.
Again, as we have already stated above, we have a completely different opinion. We believe we provided enough evidence that a new mechanism, which unfortunately was not known at the time of LP04, brings the situation regarding modeling of 4.3 μm night-time radiance at a qualitatively new level (see also discussion below).

*Page 2. Par. from lines 3 to 13. I would not argue as a motivation for this research
its potential use for retrieving CO2 from night-time 4.3 μm SABER measurements.*

The reviewer is obviously in a different situation than most of us are. Some co-authors of this manuscript are directly funded from the SABER/TIMED experiment and, therefore, it is our job to search for new mechanisms, which can improve the interpretation SABER measurements. We consider this study as an important first step toward developing new algorithms for CO2 retrieval from the night-time 4.3 μm SABER signals.

*I think the new mechanism is already important on its own, i.e., it is important to under-
stand the non-LTE processes occurring in the middle atmosphere, ….*

We agree with the referee that the new mechanism is important.

*… and it does not need the motivation of CO2 night-time retrieval because this presents additional problems, which, in my view, are more important. First, I think to measure CO2 at night-time is not very important as far as we have daytime measurements for the wanted latitudes/seasons. Because, as it is very well mentioned in the manuscript, CO2 has a very long chemical lifetime, we do not expect significant (photochemical) diurnal variations. Only tides, but they would also be present in night-time observations.*

The referee suppose that knowing daytime CO2 guarantees its accurate prediction for night-time, or makes the night-time retrievals not important and, therefore, not needed. However, this is just a guess of this referee, which is not confirmed by any observations. However, Nature is continuously demonstrating that it is much more inventive than our guesses, therefore, it may easily happen that nighttime CO2 is different, and, since CO2 (so far, that of WACCM) is used for temperature retrievals from the 15 μm emission, then temperature/pressure will be also different from those predicted by the model, and so on. Additionally, does the referee suggest to just simply forget about significant fraction of SABER observations collected during the last 15 years?

*The only region of interest would be the polar winter, where no daytime measurements are available yet. But the retrieval of CO2 there has other problems. As it is in high latitudes, auroral excitation of N2(1) is very important and that is not well known. Also the geomagnetic conditions are very variable and hence difficult to model. In addition, as has been demonstrated by Winick et al. (1988), the location of the aurora along the LOS has to be known very well. Furthermore, most of the night-time 4.3 μm radiance comes from the strong CO2(v3) fundamental band, which is very optically thick.*

The reviewer continues here with his/her strange way of thinking: the polar night is the complex region, and what? Should we better forget about it and find another object of research?

*And last, the night-time 4.3μm signal is usually much more noisy (~a factor for 100 or larger) than the daytime one.*

We agree with the reviewer regarding the reduction in the absolute value of the nighttime CO2 radiances relative to the daytime (~100 smaller). However, the NER for the SABER Ch7 is ~ 1e-6 W/M2/Sr, (see, for instance, Mertens et al, 2003). However, the nighttime Ch7 radiances are still roughly a factor of 5 higher than NER for large number of scans up to 115-120 km, see the plots at the end of this document, which show fractions of our analysis of the Ch7 signals (in W/M2/Sr), version 2.0, versus solar zenith angle. There are many profiles that have SNR >100 at 115 km, which is an interesting effect on its own, but also implies the retrieval could proceed to 120-130 km. It also implies that the nighttime retrieval will have a moving upper boundary, but it may proceed without issues at least up to 100-110km, and in some instances up to 120-130 km.

*Page 2. Line 26 and ff. "However, using laboratory rate coefficients of corresponding reactions the authors were unable to reproduce the 4.3 μm radiance observed by SABER." This is only partially correct, as they were able to reproduce SABER radiances when using an efficiency of 2.8-3 with the same reaction rate.*

This is another return to the same idea about equal importance of both new and old mechanisms. Please see our replies above (and also below).

*Reaction R2 has been normally used as OH(v,<=10) + N2(0) <=> OH(v-1) + N2 note
that N2 is not excited, see, e.g. Adler-Golden et al (1997), because it has been used
in OH(v) modelling and the interest was the deactivation of OH(v), without paying any
attention to the final state of N2, e.g., if it was excited or not excited. Hence, I think the
statement (line 30) that "its has been accepted with a value of 1" needs more discus-
sion. It has been used most of the times regardless of the excitation of N2. Theoretical
estimates by Adler-Golden et al. (1997) and Sharma et al. (2015) suggest that it takes
place at single-quantum relaxation. However, to my knowledge, the efficiency of this
reaction has not been measured in the laboratory, mainly because the major interest
was to know the relaxation of OH(v) and not where the energy goes. Are these reasons
enough for completely disregarding the multi-quantum? I do not think so.*

This is again the same topic. Please see more detail discussion below. Briefly: we do not completely disregard the "old mechanism," but we cannot accept it (with a thus far unexplained arbitrary efficiency of ~ 3) as an equivalent alternative to the new mechanism. Nevertheless, we changed the text from "instead of the accepted value of 1" to "instead of the currently accepted value of 1"

*The mechanism the authors propose sounds plausible but one should be careful about assuring
that it is "the" mechanism (and reject the LP04 mechanism). If still the authors would
like to be categorical, I think this point needs to be discussed deeper in the manuscript.
Page 3.*

We are not "categorical" at all, although we could be more categorical relying on our justifications: (a) the efficiency of the new mechanism is justified with detailed theoretical and laboratory analysis by Sharma et al 2015, and Kalogerakis et al, 2016, and (b) our study shows that the new mechanism leaves little space for other "alternatives".

*Minor comment. Lines 1-2. The proposed new mechanism strictly refers to OH(v) to
N2(1), rather than OH(v) to CO2(v3).*

We made the corresponding correction in the text.

*Lines 5-6. "Kalogerakis et al. (2016) provided a definitive laboratory confirmation for
the validity of this new mechanism." Were they able to measure the reaction rate and
energy efficiency of this mechanism? Is this new mechanism still based on the "theo-
retical" calculations of Sharma et al. (2015) for the reaction rate of the OH(v)+O(3P)=>
OH(v')+O(1D) ?*

Please see detailed discussion of the same comment below

*Lines 7-8. If the author would like to be consistent with the model of LP04 they should
use the efficiency of 2.8-3.*

Once again, we state here that we do not study the ability of the Kumer mechanism to fit SABER measurements testing a hypothetical higher efficiency. Our goal is different – to show the effect of a

newly discovered and validated mechanism when it works together with that of Kumer, which is used with the currently accepted efficiency of f=1.

*Line 8. "... OH(v) energy transfer to "N2(1)" instead of to "CO2".*

We changed the text.

*Lines 19. Again, in order to be consistent with LP04 the authors should use an efficiency of 2.8-3.*

Please see our replies above.

*Lines 25-27. The SABER data version should be stated.*

We used 2.0 version and added this information to the text

*Line 27. As the authors mentioned above CO2 has been retrieved. I was then expecting to use the retrieved CO2 instead of that of WACCM. This should give better simulated radiances and remove some uncertainties.*

What kind of "some uncertainties" does the referee mean? Our main conclusion is quite certain – the new mechanism is important and must be accounted for. Using retrieved daytime CO2 (which, meanwhile, is still not publicly available) will not change this conclusion.

*Reactions R1-R4 are repeated in the text and in the Table. Maybe they should be kept only in the Table.*

We would prefer to show reactions in both text and table. It will be easier for readers not to be referred to the table multiple times when the various mechanisms are discussed.

*Page 4.*
*Sec. 2.2*
*A major comment. As the new proposed mechanism affects also to the population of OH(v) and the emissions from these levels were measured by SABER in two different channels, I think it is essential that the authors demonstrate that the new OH(v) model explain very well the measured SABER OH radiances, as LP04 did. Thus, figures should be presented for different conditions comparing SABER observations and modelled radiances for the two OH SABER channels.*

The referee is raising a good point that we considered in our consistency checks by evaluating the effect on the OH(v) population for the two highest vibrational levels. We found very good agreement (within a few percent) between the number density determined for OH(8+9) from our test data sets and the SABER channel 8 observations. The absolute number density for OH(8+9) can be directly determined from the SABER Channel 8 radiances (Mast et al., 2013) and is therefore a rigorous consistency check. The extracted OH(8+9) absolute number density does not depend on any previous model result (only the radiance profile and the almost equal Einstein emission coefficients A97 and A86 are needed). Our ultimate goal is to develop an updated model that handles simultaneously all the SABER OH(v) and CO2 emission channels, but this is clearly out of the scope of this report. The goal of our study was to estimate, with the help of model calculations that are based on reliable inputs, the

effect of the recently discovered "indirect" pumping mechanism of N2(v) at nighttime. The latter was suggested and then experimentally confirmed by Sharma et al, 2015 and Kalogerakis et al, 2016, respectively. We believe we demonstrated the importance of this new mechanism and made significant contributions that complement the previous studies. Nevertheless, as we make clear in our conclusion statements, there is plenty of room for further research.

*About the O(3P) abundance and the OH(v) model, the authors state that they used the O(3P) retrieved from SABER measurements. The SABER O(3P) is derived from the SABER OH radiances but a photochemical OH(v) model is required for such inversion (Mlynczak et al., 2013). How do the reaction rates for the OH(v) model used here (Table 1) compare to those of Mlynczak et al., 2013? Actually, to be consistent, it should be used the same photochemical OH(v) model in both cases.*

We took as much information as possible (pressure, temperature, O, O3, H) from SABER retrievals. However total OH density was taken from WACCM to calculate OH(v) following the model similar to that described by Xu et al (2012). We repeat again here that our goal was quite limited – to estimate with the help of reliable model calculations the effect of the new N2 pumping mechanism at nighttime.

*Along this line, the mechanism proposed by LP04 did not affect the established OH(v) model (e.g. Adler-Golden et al., 1997), so in that sense it was also compatible with most of previous OH(v) emission rocket measurements. How does the new OH(v) photochemical model compare to that of Adler-Golden et al.? I.e., it is also compatible with previous OH(v) emission rocket measurements?*

Please see our comments above about OH(v) comparisons.

*Line 9. The text in this line is repeated a few lines below.*

We changed the text.

*Lines 14-15. Could the authors be more precise with "lower" and "higher" CO2 vibrational levels?*

In the majority of cases the rate coefficients are measured/calculated only for transitions between ground and first excited vibrational levels (in some cases between lower nearby lying vibrational levels). Shved at al 1998 suggested scaling these values to similar transitions between higher vibrational levels. We applied these scaling rules in our CO2 non-LTE model.

*Line 21. Then the values used for the reaction rate of the new mechanism are based on theoretical estimations? not measured values? Kalogerakis et al. (2016) did not measure those the reaction rates and efficiencies? If measured, why not use the measurements with their errors instead of theoretical estimates?*

The estimate for the total removal rate constant for OH(v = 9) + O at mesospheric temperatures, 3e-10 cm3 s-1, is based on laboratory experiments conducted at SRI International: Kalogerakis et al. (2011) investigated OH(v = 9) + O at room temperature and Thiebaud et al. (2010) reported the

temperature dependence of OH(v = 7) + O from room temperature to mesospheric temperatures, assuming that these processes have similar temperature dependence. Sharma et al. (2015) estimated the rate constant for the OH(v = 9) + O multi-quantum vibrational relaxation pathway leading to O(1D) excitation by subtracting from the total removal rate constant the contributions of inelastic vibrational relaxation and reaction to H + O2, as calculated by Varandas (2004). The corresponding rate constant estimates are 3.2e-10 cm3 s-1 and 2.3e-10 cm3 s-1 at room temperature and at temperatures near 200 K, respectively. The laboratory experiments by Kalogerakis et al. (2016) confirmed the prediction of Sharma et al. (2015) at room temperature. Therefore, the rescaled rate constant for the multi-quantum vibrational relaxation pathway used in the modeling calculations reported here also relies on the experimentally measured temperature dependence for OH(v = 7) removal by O atoms. Rescaling of available OH(v) + M rate constant values is common practice when measurements at the temperature of interest are not available in the literature (e.g., Adler-Golden, 1997; LP04). The reviewer can find more detailed discussion in the papers by Sharma et al, 2015 and Kalogerakis et al, 2016. We also changed the manuscript text to refer readers to both papers.

*Page 5. Sec. 3.2 Lines 20-25. This section and Fig. 2, when the authors refer to the calculations of LP04 with the "direct" mechanism, can be misleading. LP04 were able to reproduce the observed SABER radiances when using this mechanism but with an efficiency of 2.8.*

To avoid possible misunderstanding, we changed here the text from "... (R1-R3) only (``direct" mechanism)" to "..(R1-R3) only ("direct" mechanism with currently accepted efficiency 1)"

*"... inputs identical to those of Lopez-Puertas et al (2004)." Lopez-Puertas et al (2004) used version 1.03 of SABER parameters. Which version has been used here? Are they really identical? To which degree?*

We used version 2.0. Version 1.03 which was used by LP04, is no more available, therefore, it was impossible to rigorously compare LP04 and our model inputs. That was not our goal, to literally reproduce the LP04 results, which we show just for comparison, however, we mention that both simulated signals (for the currently accepted efficiency of Kumer mechanism) are very close. These two signals, as well as other simulated signals are compared to the measured signal from the current 2.0 version. Digitizing the Ch7 signal shown by LP04 and comparing it with the signal currently available for the same scan we found a small difference reaching maximum of 10% around 70 km, which are not important for comparisons discussed in our paper.

*Lines 25 and ff. Using OH densities from WACCM. I believe the authors mean OH(v) densities, i.e. vibrationally excited OH, not OH in the ground state. In this case, the WACCM OH(v) densities might be largely under-predicted since it is well known that WACCM mesospheric O3 abundance is underestimated by at least a factor of 2 with respect to satellite measurements (both SABER and MIPAS) (Smith et al., 2013). By the way, the authors describe the OH(v) photochemical and the sources of some atmospheric constituents but not the source of O3 and H. Or was it included OH(v) (instead of OH) from WACCM and the OH photochemical model described (Table 1) is that of WACCM?*

In our reply to RC1 to similar comments we stated (and also changed the manuscript text) that we used in these model calculations O3, H and O retrieved from SABER, but **total**

OH density was taken from WACCM, and OH(v) we calculated with our own non-LTE model.

*Page 7. Conclusions.*
*Lines 16-17: "This significant improvement suggests that the missing night-time mechanism of CO2(v3) pumping has finally been identified. " I would not be so categorical. At least experimentally it has not yet been ruled out the possibility of multi-quantum energy transfer from OH(v) to N2(1).*

N2 is still considered as quite an in-efficient quencher of OH(v), see discussions by Kumer et al 1978, Burtt and Sharma, 2008 and Sharma et al, 2015, and references therein. It has never been shown and is still not expected that a process of the type OH(v) + N2(0) --> OH(v-n) + N2(k), where n and k > 1, would have any significant probability. Additionally, (a) the verb "suggests" we used is certainly not categorical. Moreover, our confidence is based on the fact that new mechanism accounts for most of the discrepancy, thus it appears there is little room for other processes (that, of course, cannot be excluded, but are not expected to be significant), (b) everything else that follows in our conclusions is rather measured and does not claim that we know everything: "Further improvements will require optimizing the set of rate coefficients used for OH(v) relaxation by O(3P) and O2 at mesospheric temperatures and, in particular, understanding the dependence of the indirect mechanism on the OH vibrational level. Relevant laboratory measurements and theoretical calculations are sorely needed to understand these relaxation rates and the quantitative details of the applicable mechanistic pathways."

*"Relevant laboratory measurements and theoretical calculations are sorely needed to understand these relaxation rates and the quantitative details of the applicable mechanistic pathways." I understood from this manuscript that these have been already done (e.g. Sharma et al., 2015 and Kalogerakis et al. (2016). What new is needed?*

Please see the reply to previous comment.

*"Nevertheless, results presented here clearly demonstrate significant progress in understanding the generation mechanisms of the night-time CO2 4.3 µm emission and represent an important step towards developing the algorithm(s) suitable for retrieving CO2 densities in the MLT from the SABER night-time limb radiances." I agree with the first part of the sentence.*

We appreciate this at least "partial" agreement with our conclusions.

*However, I see no real progress for an eventual retrieval of CO2 from night-time radiances. SABER measurements were already reproduced before as good as with this new mechanism.*

Summarizing all discussion above, we once again draw the referee's attention to the following:

- LP04 showed that Kumer mechanism with f=1 in no way reproduces measurements. Even for very carefully selected scans the differences reached 40%

- LP04 also showed that f=~3 (possible multi-quantum process, but mechanism is not explained) removes about 40% of differences for some selected scans studied. In our very extensive study we found that these differences can reach as high as 80%. Following the LP04 logic it would

require f=6 and higher to remove these differences, which is absolutely unrealistic. In other words, LP04 quantified the energy transfer efficiency that would be required for model calculations and observations to agree (for some limited set of scans), but no detailed mechanism was described, let alone validated by referring to theoretical or experimental investigations.

- Sharma et al 2015 in a detailed analysis suggested a new efficient mechanism of transferring OH(v) energy to N2(v), and Kalogerakis et al 2016 demonstrated the validity of this hypothesis in laboratory experiments. As we showed in our extensive study the new Sharma mechanism (combined with that of Kumer for f=1) very efficiently removed up to 80% of signal differences for the large variety of atmospheric situations.

- The new mechanism leaves little room for other processes (that, of course, cannot be excluded, but are not expected to be significant at this point, and will need to be validated first theoretically or experimentally, or better yet in both ways, just as in the case of the Sharma mechanism).

- This referee agrees that Sharma's et al., Kalogerakis' et al., and our results "*demonstrate significant progress in understanding the generation mechanisms of the night-time CO2 4.3 µm emission*"

- Progress in understating observed phenomena is usually followed by practical or technical application of this new understanding, which, in our case, will be the retrieval algorithm. We invite the referee and his/her group to make their own research contributions, given the new knowledge.

*Further, even with such a good reproduction, the inversion of CO2 from night-time 4.3 µm emission in the regions where it would be useful (winter polar night) is still very difficult due to the reasons mentioned above.*

Please see our reply to the same comment above.

References

Burtt, K. D. and Sharma, R. D.: Near-resonant energy transfer from
vibrationally excited OH to N2, J. Chem. Phys., 128, 124311,
doi:10.1063/1.2884343, 2008.

Mast, J., M. G. Mlynczak, L. A. Hunt, B. T. Marshall, C. J. Mertens, J. M. Russell III, R. E. Thompson, and L. L. Gordley (2013), Absolute concentrations
of highly vibrationally excited OH($\tilde{o}$ = 9 + 8) in the mesopause region derived from the
TIMED/SABER instrument, Geophys. Res.
Lett., 40, 646–650, doi:10.1002/grl.50167.

2002, 121-125

[Figure]

**Jan 1-15. 2003**

---

## Author Response (AR2)

Dear Franz-Josef,

please see below our replies to referees' reports.

We marked there precisely the lines in the revised manuscript where we address each particular comment.

In the revised manuscript edited or new text is given in bold. This includes three new sub-sections as well as those sections from the previous manuscript version, where edited text exceeded 70%. Since the manuscript is prepared in latex it was technically not possible to mark any new word or any small sentence part changed.

With warmest regards,
Alex Kutepov

**Replies to the report of Anonymous Referee #3**

We thank this reviewer for  comments to your manuscript. Below these comments are given in italic followed by our replices.

*The authors have addressed my comments and suggestions but very little changes have been taken into account in the revised version, eluding the most important calculations. Also, I do not like the tone of the response, e.g. sentences like "We suggest the referee read more carefully the LP04 paper ..." or "... his/her strange way of thinking..." or "We invite the referee and his/her group to make their own research contributions, given the new knowledge" are completely out of place and should be omitted. That said, I have tried to be as objective as possible.*

We are very sorry if our replies to this reviewer comments looked inappropriate.

*I think the paper deserves to be published. I am making the less possible comments and are divided in: a) essentials (without which I would not recommend the publication) and b) important (highly recommendable). I also include a comment on the authors' comment at the end that do not affect the manuscript but, since this report will be public, I feel I should reply.*

We changed the title and the text of paper to satisfy all *essential* and *important* comments of this reviewer except of comments 4 and 5. We believe that the requirements in Comment 5 are in contradiction with those in Comment 4 (please see detailed replies to comments 4 and 5 below).

*One of the major points comes from the interpretation of the sentence in LP04 of "...We have investigated the SABER 4.3 μm radiances with the help of a non-LTE radiative transfer model for CO2 and found that the large radiances can be explained by a fast and efficient energy transfer rate from OH(v) to N2(1) to CO2(v3), whereby, on average, 2.8–3 N2(1) vibrational quanta are excited after quenching of one OH(v) molecule." We may question if this mechanism is realistic or not but it is a mechanism and LP04 were able to reproduce the radiances for essentially all conditions (e.g. for 4 orbits of 4 days covering equinox and solstice (Figs. 12-14)) within +/-20%, that is, equally or better than with the new mechanism.*

In the new manuscript version we removed any discussions of which model and how well it **fits**

SABER measurements. The paper deals now only with the comparison of various model calculations using WACCM self-consistent inputs of all atmospheric parameters. Although SABER measures emissions in three channels, and now also ground and space observations of OH(v) distributions are involved in the discussion, they are given only as reference data to illustrate how well various model reproduce basic features of measurements.

*True that so far there is no evidence of such a "direct" energy transfer process and this is why this manuscript is relevant, because it gives a plausible "indirect" way of such an energy transfer, even though the actual reaction rates have not been measured (only estimated) and the efficiency of OH(v) to O(1D) has not been measured either. Hence, I do appreciate this qualitative new mechanism.*

We are puzzled with this statement. The comment 4 below shows, that reviewer is aware of the works by Sharma et al 2015 and Kalogerakis et al 2016. In the latter paper reaction rates for both OH(v)+O3P and OH(v)+O1D were measured. We use results of this measurement in our calculations.

*That is why I suggested to focus (title, abstract, etc.) on the new pathway more than on if it is able to reproduce better or not the measured SABER $CO_2$ 4.3 µm radiances. Then, my comments:*

We followed this suggestion and show now the work of new pathway in comparison with other models. Therefore, we show observation results only for illustration of how well various models reproduce basic features of different measurements.

*1) Title: (important). I would still recommend changing the part "New model calculations improve agreement with SABER observations".*

We changed the second part of the title. It is now neutral saying only that we compared various models with observations: **Comparison of the CO2(v3) and OH(v) emission models with space and ground based observations**

*2) Abstract (important): If with the "previous study" the authors refer to "Kumer et al." it is correct. However if to LP04, it is also correct but not complete (see above). My recommendation would be to give the whole story not only part of it.*

It is barely possible to tell the whole story of previous studies in the abstract. We do this in Introduction. However, both in Abstract (lines 1-5) and Introduction (lines 16-23) we underscore an importance of the *LP04* study who, for the first time, quantified required efficiency of the OH(v) energy transfer to N2 to fit 4.3 µm observations.

*3) Motivation of the work (clarification, not relevant for the manuscript). I tried to say that this work is important itself and does not need the additional justification of retrieving CO2 at night. Of course, in no way I meant to discourage the authors in pursuing such research.*

We removed from the text any mentioning of our motivation for this study. The paper deals now only with comparison of various models.

*4) (Essential). Since there have been already two papers dealing with this new mechanism (Sharma et al. (2015) and Kalogerakis et al. (2016)), I think this work should make the most accurate and*

*consistent calculations as possible, making use all available SABER data in a consistent and proper way. That it, in my opinion is not valid the argument that the purpose of this work is to make "estimates" and further work will be done later. Hence:*

This comments is addressed below together with comment 5)

*4a) They should use the retrieved CO2 from SABER (contrary to their reply, CO2 is publically available, (*ftp://saber.gats-inc.com */Version2_0/Level2C/) and two of the co-authors are co-authors of the CO2 retrieval papers (Rezac et al., 2015a,b).*

We would like to remind here that Rezac et al, 2015 retrievals were performed only for daytime SABER observations. In this study we modeled nighttime conditions. Additionally, retrievals of Rezac et al, 2015 use the SABER retrieved O3P which is considered to be too high compared to other observations (see also comment 5). To avoid a risk of using inconsistent inputs we switched to WACCM based inputs.

*4b) My previous major comment on the OH SABER radiances (see below) has not been addressed adequately. "As the new proposed mechanism affects also to the population of OH(v) and the emissions from these levels were measured SABER in two different channels, I think it is essential that the authors demonstrate that the new OH(v) model explain very well the measured SABER OH radiances, as LP04 did. Thus, figures for different conditions with the SABER observations and modelled radiances for the two OH SABER channels should be presented in this work."*

We followed this request and show now comparison of our model with SABER OH(v) emissions (new Figure 3, lower raw, the discussion is given in the new Section 3.4 " **The OH 1.6 and 2.0 um emissions").**

*Thus, their replies of: "... but this is clearly out of the scope of this report." or "The goal of our study was to estimate, …" I think it is crucial for this manuscript (not report), and I think it should be something more than an estimate, the title of the work reads "New model calculations improve agreement with SABER observations"*

We are not sure we understand this recommendation: in one of the previous comments we were suggested to change the title, while here it is suggested to improve the paper to make it more consistent with the original title.
As we replied above, the title was changed to be consistent with the revised paper which deals exceptionally (as it was suggested by this reviewer) with comparing new mechanism with other models.

*5) (Essential, in the same line as point 4). Atomic oxygen is key for the new excitation mechanism, therefore all models inputs and SABER radiances should be consistent. Thus, my previous point on this topic has not been adequately addressed. Taking from my previous report:*
*"About the O(3P) abundance and the OH(v) model, the authors state that they used the O(3P) retrieved from SABER measurements. The SABER O(3P) is derived from the SABER OH radiances but a photochemical OH(v) model is required for such inversion (Mlynczak et al., 2013)." The authors should use the same photochemical OH(v) model or prove (with calculations and figures) that they are consistent. Further on this topic, several works have shown that SABER atomic oxygen might be overestimated in a ~30%. It would very important to comment, how would this affect to the simulations of SABER CO2 4.3 µm nighttime radiances with this new mechanism?*

This comment is in contradiction to comment 4, which requires to "*make the **most accurate and consistent calculations** as possible, **making use all(!) available SABER data in a consistent and proper way.**" On the other hand, comment 5 is about an internal inconsistency of SABER retrieval products: O3P is supposed to be too high, indicating that the OH(v) model currently applied for its operational retrieval may require update. In this study, we apply new research for the OH(v) model, which utilizes new OH(v)+O3P mechanism. The latter is missing in the current operational model. Therefore, using any SABER inputs to test this new model would be not logical. To be on a safe side, we show the calculations based exceptionally on inputs for night-time atmospheres obtained from the WACCM model.

*Further on this topic, several works have shown that SABER atomic oxygen might be overestimated in a ~30%. It would very important to comment, how would this affect to the simulations of SABER CO2 4.3 μm nighttime radiances with this new mechanism?*

Actually, this topic was discussed in the initial version of this manuscript. Since results we present now are based exceptionally on the WACCM inputs we do not address this point.

*6) Sec. 3.2 and Fig. 2 (Essential). I cannot see the reasoning of why using only the partial result of LP04. Do the authors want to validate their model? I see a high risk to misleading the reader as giving the impression that LP04 were not able to explain the SABER radiances when they did (see Figs. 10 and 11 in LP04). I strongly recommend to either show the two LP04 simulations or none.*

In the revised version of Fig.2 we show how well we reproduce both initial and final (with the 3 times higher efficiency) results of LP04. We discuss in detail **both** results and compare them with other models and measurements:
- in new section 2.3, where we describe in detail all models we used in our calculations, p4, lines 25-33, p5, lines 11-13,
- in section 3.1, p5, lines 26-29, p5, lines 1-5
- in section 3.2 which discusses revised Fig.2 and 3, nearly a half of text is about results of LP04 and their comparison with other models.
- In Conclusion p10, lines 30-34

In all these discussions we stress several times that LP04 were able to reproduce SABER 4.3 um measurements with their final model as good as we do it with our new model based on the new "indirect" mechanism.

*7) (Essential) About the OH densities. If the authors calculate OH(v) (does this include also v=0?) from SABER O3 and H, why they need OH (ground state? or total (i.e. OH(v) including v=0) from WACCM? If this is important, my previous argument whereby WACCM should sub-estimate OH still applies. As mentioned above, the authors should use a consistent model and inputs for all quantities.*

As it was already discussed above we use in a new version of manuscript exceptionally WACCM inputs.

*8) (important) Conclusions. As mentioned at the beginning I would focus more on the mechanism itself (find the way to transfer so much energy from OH(v) to N2(v) rather than on the "improve agreement" of the SABER radiances.*

Again, following this recommendation we focused now on the study of new mechanism and its comparison with other models of OH(v) relaxation. However, this theoretical study without any comparison with reference measurements would be quite useless.

Therefore, we show SABER and other measurements, compare calculations with them, and discuss how various models reproduce main features of observations.

*I do not fully agree with your statement that "(b) everything else that follows in our conclusions is rather measured ..." You need to make "estimates" of key parameters as how much energy is transferred from OH(v) to O(1D), both on the collisional rates and their temperature dependences.*

This comment is appropriate (as a short note publication, for instance) to the papers by Sharma et al, 2015, and Kalogerakis et al 2016, where "estimates" of these key parameters was performed. It can be hardly addressed to this work where we use results of these studies to show how strong they may influence the 4.3 um emission modeling.

*Just one clarification to your statement:*
*• LP04 also showed that f=~3 (possible multi-quantum process, but mechanism is not explained) removes about 40% of differences for some selected scans studied. In our very extensive study we found that these differences can reach as high as 80%. Following the LP04 logic it would require f=6 and higher to remove these differences, which is absolutely unrealistic. In other words, LP04 quantified the energy transfer efficiency that would be required for model calculations and observations to agree (for some limited set of scans), but no detailed mechanism was described, let alone validated by referring to theoretical or experimental investigations. This is not fully true. LP04 showed, not only for some limited scans but for 4 orbits of 4 different days covering equinox and solstice conditions, that SABER measurement with f=2.8-3 could be reproduced within +/-20% (see Figs. 12-14). I would not be surprised to need larger f-values for the polar regions, where auroral excitation is prone and frequent. Also, if you analysed so many data, it would be very useful to the reader to show more than just only 2 days (Fig. 3).*

In the new version we removed any discussion regarding **fitting** SABER measurements, what efficiency needed for this, etc. The paper deals now only with comparison of various models using WACCM self-consistent inputs of all atmospheric parameters. Although SABER measurements in three channels, as well now also ground and space observations of OH(v) distributions are involved in the discussion, they are given only as reference data to discuss how vell different models reproduce main observational features.

**Reply to the report of Anonymous Referee #2**

Below the reviewer's comment is reproduced in italic and is followed by our reply.

*The authors have addressed satisfactorily most of my comments, however, I still think that the paper would benefit substantially from a figure that shows the modeled and observed OH(v=8,9) radiances (SABER channel 8), similarly as Fig. 2 does for CO2 4.3 um (SABER channel 7). The point is that with the proposed mechanism the excitation of CO2 is ruled by the number density of the excited OH states and hence a rigorous test would be to compare both OH\* and CO2 radiances with the SABER observations. The authors state in their reply that such a consistency test has already been performed, so why not including it in the manuscript?*

We are very grateful to this reviewer for his friendly comments, which helped to improve this manuscript.

In revised version of manuscript we completely accounted for suggestions above and compare in new Fig.3 both CO2 and OH emission calculations with SABER measurements. Detailed discussion of these results is given

- in p.7, lines 13-23 (CO2 emissions)
- in new section 3.4 p.9, lines 20-33 and p.10, lines 1-12. (OH emissions)

---

## Author Response (AR3)

Dear Professor Lübken,

please see below our replies to the referee report. We marked in bold the modified manuscript text.

With warmest regards,
Peter Panka

*The revised manuscript includes important new analysis relevant to the topic. However, I'm puzzled about the authors' reply to Reviewer#3 regarding the focus of the paper. They say that they "removed any discussions of which model and how well it fits SABER measurements" , dealing now "only with the comparison of various model calculations using WACCM self-consistent inputs of all atmospheric parameters". But then the new title reads "...:Comparison of the CO2(v3) and OH(v) emission models with space and ground based observations". Further, they conclude, e.g., that the new indirect channel provides significant 4.3 um emission enhancement strong enough to \*FIT\* SABER radiances. This seems not consistent to me with their statement in their reply.*

We thank the referee for pointing out the remaining inconsistencies. We changed the title to make it clear that our main goal is the comparison of models. The manuscript text was also changed correspondingly.

*If the focus of the paper was limited to an intercomparison of various models then, in consequence, any comparison to observations should be excluded and the manuscript title and some parts of the manuscript need to be rewritten. In this case, it would be most appropriate to use the (OH-N2 3Q) & (OH-O2 1Q) & R7 model as reference in the comparisons of simulated 4.3 um emissions because its capacity to reproduce SABER observations (4.3, 2.0, and 1.6 um) in a self-consistent manner has been demonstrated in a previous study.*

As we have pointed out previously, in the revised paper we focused primarily on the comparison of various models. However, the final goal of any modeling is the interpretation of measurements, we still believe that there is a compelling reason to show comparisons not only among models, but also with various measurements. These comparisons enable us to determine how well each model reproduces basic observed features and thus shed light to the physical processes we are attempting to explain. For instance, we are puzzle by the argument against us showing the comparison between a published model such as the one presented in Lopez-Puertas et al, 2004 and the measurements the model attempts to reproduce. These are published results and for complete scientific sake must be addressed when new results show potential disagreements.

Additionally, we disagree with the reviewer in that we should remove the comparison of models with ground and space observations of OH(v) distributions (Figure 4). This is the compelling point of our study. Without such comparison, the modeled results would represent just an unconstrained theoretical exercise.

*However, If they aim at evaluating the various models with observations (as suggested by the present title) it is absolutely necessary to also demonstrate that model inputs and intermediate results are consistent with the measurements. In particular, it then needs to be demonstrated that modeled ABSOLUTE OH\* densities are consistent with SABER 1.6 and 2.0 um measurements. This was already*

*requested by both reviewers in the previous iteration. However, the authors now show RELATIVE populations or VER ratios in comparison with observations. This is a very interesting additional diagnostics, however, it does not allow to judge if one or the other model provides a better description of energy transfer from OH(v>4) to CO2 because the amount of available excited OH is not constrained. In other words, good agreement of modeled and 4.3 um emissions could also be achieved by a combination of a wrong energy transfer mechanism in combination with wrong OH\* densities.*

In this study we do not fit any measurements, however, show (on fixed inputs) how various models reproduce specifics of **various measurements**. The total OH density in our calculations is fixed by the WACCM inputs. We show here how various relaxation mechanisms **impact the distribution of the OH(v) vibrational level populations and CO2 pumping**. Regarding the referee's statement that our model *"does not allow to judge if one or the other model provides a better description of energy transfer from OH(v>4) to CO2 because the amount of available excited OH is not constrained":* in our study, where total OH is constrained with WACCM inputs, we show in detail that the new mechanism provides the same enhancement of the 4.3 um emission as the best (Lopez-Puertas et al 2004) model does, even if efficient multi-quantum quenching by O2 (reaction R6) is applied (see also discussion below). Thus we still believe our current comparisons are justified and valid.

*This is a major concern, however, it can be easily addressed, either by some minor changes to the text (first case) or by inclusion of an additional row in Fig 3 showing observed and modeled 1.6+2.0 um ABSOLUTE VERs (second case). If in the latter case disagreement between modeled and observed VERs is obtained, a scaling of the WACCM [H]\*[O3] and consequently [O] (because of chemical equilibrium) could be applied in order to achieve agreement.*

In addition to our comments in the previous point, our work represents an 'in between' state of the two scenarios described by the reviewer. We do not fit any signals with WACCM inputs, therefore, we do not see the benefits to show absolute data in the framework of this study. Absolute OH VERs do not explain what is the basic difference between models, which is our main goal of the study. However the VER ratios and relative population distributions displayed in Figure 4 are the appropriate quantities to compare.

*Minor comments:*

*p5 l19-21: The use of pT from WACCM (instead from SABER as in the previous version) could introduce additional uncertainties in the comparison of modeled and observed 4.3 um emissions, which should be discussed. For instance, it is likely that the pronounced differences of both models below 75 km compared to the measurements in Fig 3 (upper panel d) are related to a temperature mismatch of WACCM. Note that, despite of being self-consistent, WACCM output does not necessarily reflect the actual atmospheric conditions location and time because the model is free-running in the mesosphere even in the SD mode. Real and modeled meteorology can thus be quite different.*

In the previous version, the reviewers criticized the use of SABER as inputs to our model. Given this criticism and the fact that we do not aim, now and then, to fit any signals, we decided to use self-consistent WACCM inputs and show measurements to only illustrate the performance of various models. We absolutely agree with this referee that compared to (inconsistent) SABER inputs, WACCM inputs *"do not necessarily reflect the actual atmospheric conditions of a given measurement"*. However, they appear to be good enough to model specifics of each type of measurements (be these SABER observations for various latitudes/seasons or ground and space observations of OH(v)

emissions) and show how well various models account for these specifics.

*It is also not clear from the provided references what kind of WACCM simulation has been used. The Marsh et al. reference points to a free-running simulation while the Solomon et al reference points to a SD simulation, however, only for the year 2011.*

For all simulations, we use the WACCM model described in Marsh et al. [2013]. We have removed the citation for Solomon et al. [2015], as this is not relevant to our study.

*p7 l9-11: There seems to be a misunderstanding of what is stated in Lopez-Puertas et al. regarding the treatment of R6 as multi-or single quantum process. The latter authors adjusted a reference OH_ref(v,z) profile that has been modeled with different implementations of R6 to the observed 1.6 and 2.0 um radiances and used then the adjusted OH(v,z) profile to simulate 4.3 um emissions. Their statement refers to the fact that changes in the vibrational distribution within v=1-5 (used to adjust the 1.6 um measurements) and that within v=6-9 (used to adjust the 2.0 um measurements), caused by the different implementations of R6, had little impact on the simulated 4.3 emission. It is evident that there would have been a significant impact if the OH(v=1-5) and OH(v=6-9) were not adjusted to the SABER 1.6 and 2.0 um radiances. Therefore, the sentence on l9-11 should be removed in order to avoid confusion.*

We thank the referee for this detailed explanation of how Lopez-Puertas et al, 2004 dealt with single- and multi-quantum implementation of reaction R6. We now understand that in order to compensate significant OH(v) decay due to multi-quantum quenching by collisions with $O_2$ and keep the transfer of energy to $CO_2$ unchanged *"OH(v=1-5) and OH(v=6-9) were adjusted to the SABER 1.6 and 2.0 um radiances"*, obviously with higher OH(v). This actually may mean that total OH density (or VMR) was increased.

We note here also that neither initial nor final *ABSOLUTE* OH(v) (or total OH densities) obtained in SABER OH signal fittings were shown and discussed by Lopez-Puertas et al, 2004. It, therefore, is not possible to judge how realistic they were, what $O_3$ and H were used, etc.

On the other hand, in this study we demonstrate, based on fixed self-consistent WACCM inputs, that the Sharma mechanism provides efficient energy transfer to $CO_2$, which as oppose to Lopez-Puertas et al, (2004), does not require additional OH adjustment to compensate the multi-quantum $O_2$ quenching. We provided additional text to make this point clearer.